# The binding and mechanism of a positive allosteric modulator of Kv3 channels

Qiansheng Liang [1,2,7], Gamma Chi [3,7], Leonardo Cirqueira [4], Lianteng Zhi[1,2], Agostino Marasco [5], Nadia Pilati[5], Martin J. Gunthorpe [6], Giuseppe Alvaro[5], Charles H. Large [6], David B. Sauer [3], Werner Treptow [4] & Manuel Covarrubias [1,2] ✉

Small-molecule modulators of diverse voltage-gated K⁺ (Kv) channels may help treat a wide range of neurological disorders. However, developing effective modulators requires understanding of their mechanism of action. We apply an orthogonal approach to elucidate the mechanism of action of an imidazoli-dinedione derivative (AUT5), a highly selective positive allosteric modulator of Kv3.1 and Kv3.2 channels. AUT5 modulation involves positive cooperativity and preferential stabilization of the open state. The cryo-EM structure of the Kv3.1/AUT5 complex at a resolution of 2.5 Å reveals four equivalent AUT5 binding sites at the extracellular inter-subunit interface between the voltage-sensing and pore domains of the channel's tetrameric assembly. Furthermore, we show that the unique extracellular turret regions of Kv3.1 and Kv3.2 essentially govern the selective positive modulation by AUT5. High-resolution apo and bound structures of Kv3.1 demonstrate how AUT5 binding promotes turret rearrangements and interactions with the voltage-sensing domain to favor the open conformation.

Voltage-gated K (Kv) channels play diverse critical roles as regulators of active electrical signaling in excitable tissues, such as the brain, heart, and muscle. For instance, Kv channels shape the repolarization of action potentials, determine the latency to the first spike in a train of action potentials and the frequency of repetitive spiking[1,2]. The molecular basis of this functional diversity resides in 40 Kv channel genes classified in 12 sub-families and many additional genes encoding ancillary beta subunits, which are differentially expressed in relevant tissues and parts of the brain and different subcellular compartments[3–6]. This diversity has stimulated a special interest in developing small-molecules and peptides that selectively modulate key aspects of function in excitable tissues and, therefore, could have potential as novel medicinal agents[7–12]. The search for Kv channel

openers or positive allosteric modulators (PAMs) has gained most of the attention because they could help treat common hyperexcitability disorders (epilepsy, neuropathic pain, tinnitus, cardiac arrhythmias, etc.) and develop new anesthetics. In contrast, negative modulation of Kv channels may help treat conduction disorders, which are characterized by hypoexcitability. Presently, there is only a limited number of potentially useful Kv channel PAMs. For example, retigabine, one of the best characterized Kv channel PAMs that targets Kv7.2/Kv7.3 complexes was developed as an anticonvulsant[8,13–16]. A search for Kv3 channel-specific PAMs led to the discovery of the imidazolidinedione derivatives that are the subjects of this study[17].

Kv3 channels are high-voltage activated Kv channels that belong to the superfamily of Shaker-related Kv channels[3]. As such, Kv3

[1]Department of Neuroscience,, Sidney Kimmel Medical College of Thomas Jefferson University, Philadelphia, PA 19107, USA. [2]Jack and Vicki Farber Institute for Neuroscience and the Jefferson Synaptic Biology Center, Sidney Kimmel Medical College of Thomas Jefferson University, Philadelphia, PA 19107, USA. [3]Centre for Medicines Discovery, Nuffield Department of Medicine, University of Oxford, Roosevelt Drive, Oxford OX3 7DQ, UK. [4]Laboratorio de Biologia Teorica e Computacional, University of Brasilia, Brasilia, Brazil. [5]Autifony Srl, Istituto di Ricerca Pediatrica Citta' della Speranza, Via Corso Stati Uniti, 4f, 35127 Padua, Italy. [6]Autifony Therapeutics, Ltd, Stevenage Bioscience Catalyst, Gunnels Wood Road, Stevenage SG1 2FX, UK. [7]These authors contributed equally: Qiansheng Liang, Gamma Chi. ✉e-mail: Manuel.Covarrubias@jefferson.edu

channels are domain-swapped tetrameric assemblies, in which each subunit is characterized by three conserved regions, the regulatory cytoplasmic N-terminal T1 domain, and two membrane-spanning regions, including the voltage sensing domain (VSD) and the pore domain (PD)[18]. The VSD is composed of four segments (S1–S4), and the PD includes the selectivity filter flanked by segments S5 and S6[18]. Whereas the VSD is mainly responsible for sensing the transmembrane voltage, the PD determines K⁺ selectivity, permeation, and gating. All members of the Kv3 sub-family (Kv3.1, Kv3.2, Kv3.3, and Kv3.4) are mainly expressed in axons and nerve terminals of neurons in the neocortex, the hippocampus, the basal ganglia, the thalamus, the cerebellum, and the brain stem[19,20]. Therein, the fast-spiking phenotype of neurons depends on the expression and specialized biophysical properties of Kv3 channels[19]. Also, Kv3 channels are major determinants of action potential repolarization in neurons with diverse electrophysiological phenotypes, including the neuromuscular junction[19,21–27]. A growing number of recently discovered pathogenic Kv3 gene variants have been linked to idiopathic developmental epileptic encephalopathies (DEE), progressive myoclonus epilepsy (PME-7), intellectual disability and ataxia, which is driving a keen search for effective therapeutic interventions based on small molecules targeting these channels[28–30]. We and others have previously characterized the biophysical and pharmacological properties of the imidazolidinedione derivatives and have found that they are relatively selective PAMs of Kv3 channels[31–34]. Positive modulation of Kv3 channels has a significant impact on the neuron's ability to generate fast-spiking, with the potential to be beneficial in disorders of the auditory system, and disorders associated with cognitive deficits due to dysfunction of corticolimbic circuits, where high-frequency gamma power and network synchronization is impaired[32,35–39]. Recent studies suggest potentially beneficial effects of these compounds in progressive myoclonus epilepsy and psychotic disorders[40,41]. Therefore, understanding the mechanism of action of these compounds may help optimize their use across different disorders.

Here, we use an orthogonal approach to determine the biophysical and structural basis of the highly selective positive modulation of Kv3.1 and Kv3.2 by AUT5. The results show that AUT5 with an $EC_{50}$ of 3.2 μM induces cooperative positive modulation by preferentially stabilizing the open state. High-resolution cryo-EM revealed the AUT5 binding site location in Kv3.1 at the extracellular inter-subunit interface between the VSD and PD and near the unique turret region of the PD. This binding site was conclusively validated by blind and focused docking calculations and additionally solving the cryo-EM structure of Kv3.1 bound to AUT1, a derivative of AUT5 with similar structure and properties, albeit exhibiting lower potency. Structural, computational, and mutational analyses, along with functional validation, also revealed that the specific turret region sequence determines the specificity of the positive modulation and that the transduction mechanism underlying the stabilization of the open conformation involves direct and allosteric interactions between the extracellular S1–S2 and S3–S4 loops and a permissive turret conformation. The insights gained here may enable further development of specific Kv3 modulators to treat a range of neurological and psychiatric disorders more effectively.

## Results

### Highly selective positive modulation of Kv3.1 and Kv3.2 by AUT5
AUT5 is a potent PAM of Kv3 channels. However, the selectivity of this modulation among diverse Kv channels has not been established. To address this, we tested several Kv channels representing phylogenetically related subfamilies upon heterologous expression in *Xenopus* oocytes and characterization of the expressed currents using two-electrode voltage-clamping (TEVC) before and after bath application of 2 μM AUT5 ("Methods"). Whereas the peak conductance – voltage ($G_p$-$V_c$) relations of Kv1.2, Kv2.1, K-Shaw2, Kv3.4 and Kv4.2 were not significantly affected, those of Kv3.1 and Kv3.2 exhibited significant changes (Supplementary Information, Figs. S1, S2). The $G_p$-$V_c$ relation of Kv3.1 was leftward shifted ($\Delta V_{0.5} = -11.2 \pm 1.0$ mV, $n = 13$, $P = 7 \times 10^{-8}$), the equivalent gating charge was slightly reduced ($\Delta z = -0.15 \pm 0.05$ $e_0$, $P = 0.011$) and the $G_{max}$ was modestly increased ($\Delta G_{max} = 8.7 \pm 1.9\%$, $P = 0.003$). The $G_p$-$V_c$ relation of Kv3.2 was also leftward shifted ($\Delta V_{0.5} = -26.5 \pm 0.9$ mV, $n = 74$, $P = 3.0 \times 10^{-42}$), and the equivalent gating charge was also modestly reduced ($\Delta z = -0.84 \pm 0.06$ $e_0$, $P = 3.4 \times 10^{-22}$). The $G_{max}$, however, was also modestly reduced, albeit the change was variable ($\Delta G_{max} = -6.0 \pm 1.1\%$, $P = 5.8 \times 10^{-6}$) (Supplementary Information, Figs. S1, S2). These effects were reversible upon washout of the compound (Fig. 1), and qualitatively like the effects of AUT1, another imidazolidinedione derivative with PAM properties but lower potency (Supplementary Information, Fig. S3). The relative insensitivity of Kv3.4 (a highly homologous Kv3 channel) to modulation by AUT5, which was confirmed in HEK293 cells (Supplementary Information, Fig. S4), was surprising and potentially significant from the mechanistic and physiological perspectives. In agreement with the differential modulation of Kv3 channels by AUT5, we found that the presence of Kv3.4 subunits dampened positive modulation by AUT5 when Kv3.1 and Kv3.4 were co-expressed to promote the assembly of Kv3.1/Kv3.4 heteromultimers that exist natively in certain neurons[19] (Supplementary Information, Fig. S5).

### Preferential stabilization of the open state and positive cooperativity underlie the positive modulation of Kv3.2 by AUT5
Since Kv3.2 exhibits the largest AUT5-induced hyperpolarizing shift of the $G_p$-$V_c$ relation, we pursued in-depth biophysical and structural characterizations of this modulation to elucidate the mechanism of action. To determine the biophysical basis of the hyperpolarizing shift, we first characterized the voltage dependence of the gating kinetics before and after bath application of 2 μM AUT5. Deactivation and activation kinetics were quantified by measuring the time constants of macroscopic tail current relaxations over a range of repolarizing voltages and the activation trajectories of depolarization-evoked macroscopic currents, respectively (Methods). Inspection of the currents suggested a substantial slowing of the tail currents at hyperpolarized voltages (Fig. 1a–e). A plot of the voltage dependence of the time constants of deactivation and activation yielded the expected bell-shaped curve with a left arm corresponding to the voltage dependence of deactivation kinetics and a right arm corresponding to the voltage dependence of activation kinetics, and a maximum that approximately aligns with the $G_p$-$V_c$ $V_{0.5}$ ("Methods"; Fig. 1f). Consistent with the analysis of $G_p$-$V_c$ relations, the time constant versus voltage curve exhibits a substantial AUT5-induced hyperpolarizing shift that is caused by preferentially increasing the time constants of deactivation (i.e., slower tail current relaxations) (Fig. 1f). For instance, the deactivation time constants at −70 mV were $1.3 \pm 1.0$ ms and $11.8 \pm 4.0$ ms in the absence and presence of 2 μM AUT5, respectively ($n = 9$, $P = 0.03$). By contrast, the time constants of current activation at voltages >10 mV are unaffected by AUT5 (Fig. 1d, e). Therefore, the AUT5-induced hyperpolarizing shift of the $G_p$-$V_c$ relation results from slowed deactivation, which indicates preferential stabilization of the Kv3.2 open conformation. Then, to quantify the concentration dependence of the positive modulation of Kv3.2 by AUT5, we created an aggregate plot of the AUT5-induced $\Delta V_{0.5}$ of the $G_p$-$V_c$ relation versus concentration of AUT5 and characterized it empirically by assuming a logistic equation (Methods) (Fig. 1g). The best fit of the logistic equation yielded $EC_{50} = 3.2$ μM and $n_H = 1.9$. This result additionally suggests that positive cooperativity involving multiple interacting binding sites underlies the preferential stabilization of the open state by AUT5.

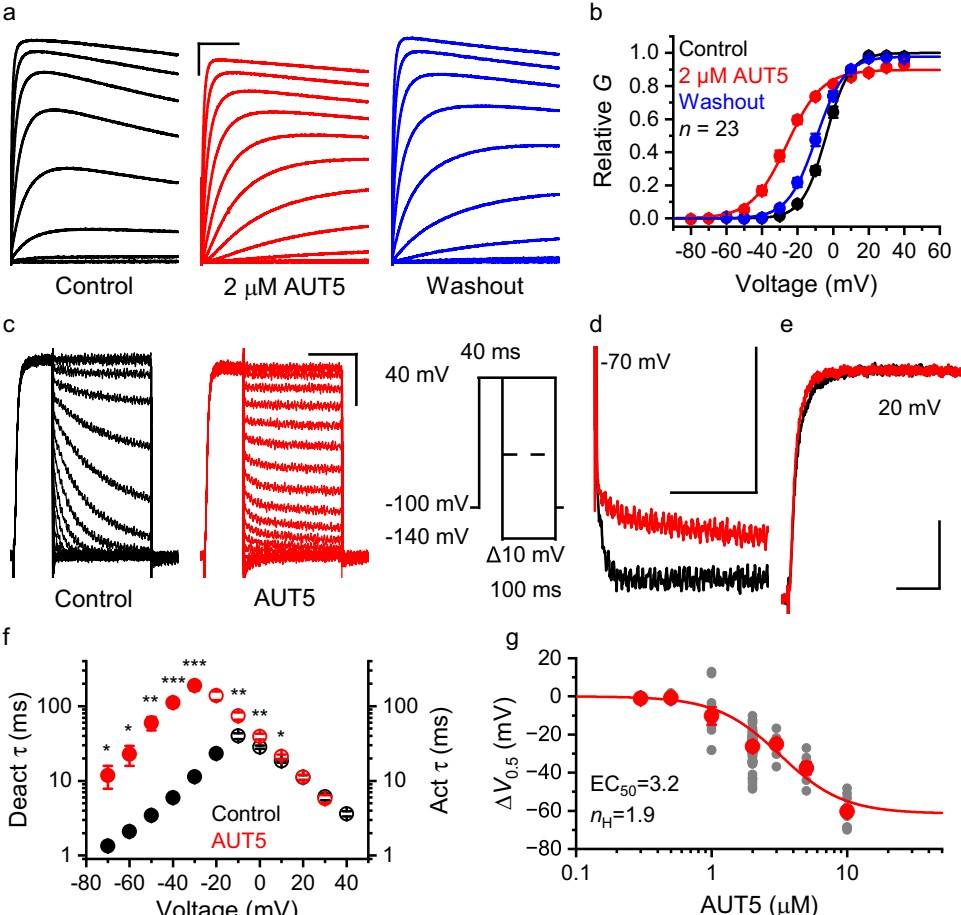

**Fig. 1 | AUT5 is a positive allosteric modulator of Kv3.2. a** Families of whole-oocyte currents evoked by step depolarizations (protocol on Fig. S1d) before and after application to 2 µM AUT5 and following washout. Scale bars represent 1 µA and 100 ms. **b** Aggregate $G_p – V_c$ curves. The solid lines are the best fits of the 1st-order Boltzmann equation (Methods). SEM bars are obscured by the symbols. The curves are normalized relative to the control in the absence of compound. $n = 23$ oocytes. The best-fit parameters are in Table S6. **c** Families of tail currents before (black) and after (red) bath application of 2 µM AUT5. Currents were evoked by the pulse protocol shown on this panel. To accurately measure slow time constants in the presence of AUT5, the tail current portion of the protocol was 1000 ms. **d** Overlay of tail currents at −70 mV in the absence (black) and presence (red) of 2 µM AUT5. **e** Overlay of currents evoked by a step depolarization from −100 to +20 mV the absence (black) and presence (red) of 2 µM AUT5. Scale bars

(**c**–**e**) represent 0.5 µA and 50 ms. **f** Voltage dependence of the time constants of current deactivation (filled symbols) and current activation (hollow symbols) before (black) and after (red) bath application of 2 µM AUT5 ($n = 9$ oocytes). At some voltages, SEM bars are obscured by the symbols. Asterisks in (**f**) indicate $P < 0.05$ (*), $P < 0.01$ (**) and $P < 0.001$ (***). $P = 0.029$, 0.015, 0.002, 7.0E-5, 1.4E-4 (−70 to −30 mV), and 0.002, 0.002, 0.027 (−10 to 10 mV) (two-sided paired Student $t$-test before and after bath application of AUT5). **g** The effect of AUT5 concentration on the $\Delta V_{0.5}$. Symbols and bars represent means ± SEM. Each filled gray symbol represents a measurement from an individual oocyte. Number of oocytes: 4 (0.3 µM), 5 (0.5 µM), 10 (1 µM), 74 (2 µM), 9 (3 µM), 13 (5 µM), and 25 (10 µM). The solid line is the best fit of the logistic equation with the parameters shown on the graph (Methods).

## High-resolution cryo-EM reveals the structure of the AUT5 binding site in Kv3.1

Structural identification of the AUT5 binding site in a Kv3 variant that exhibits selective positive modulation by AUT5 is a necessary step toward understanding the mechanism of action. Thus, considering that AUT5 is also a positive modulator of Kv3.1 (Supplementary Information, Fig. S1) and our previous work that solved the cryo-EM structure of the Kv3.1 in the apo conformation[42], we pursued the determination of the cryo-EM structure of the AUT5 bound conformation (Kv3.1/AUT5 complex) (Methods). This effort revealed the open-like structure of Kv3.1/AUT5 complex at 2.5 Å nominal resolution (Fig. 2a, Supplementary Information, Fig. S6).

The Kv3.1/AUT5 complex has an overall conformation like that of the previously determined apo condition, with the voltage sensor residues in S4 helix in the UP configuration and the internal cavity of the pore domain open to the cytoplasmic side (Supplementary Information, Figs. S7, S8). A non-protein in the electron density map consistent with AUT5 is found in a cavity between S4 and S5 helices on the

extracellular side of the channel, bounded by V312 and F315 in the S4 helix, and M362, Y365, I369 and A371 of the S5 helix (Fig. 2b). Compound binding is stabilized by polar interactions between AUT5's imidazoline-2,4-dione (ID) group and peptide backbone atoms of R368, I369 and A370, and by its spiro[2H-1-benzofuran-3,1'-cyclopropane]-4-yl (SBC) group in a hydrophobic pocket formed by F315, M362 and Y365 side chains in the middle of the bilayer interface (Fig. 2b). Interestingly, most of the AUT5-interacting residues are well-conserved between all four members of Kv3 channels but not with other Kv channels (Supplementary Information, Figs. S9 and S10), which suggests that AUT5 may retain at least some binding affinity for all Kv3 channels.

The identified interactions caused significant local changes to Kv3.1's inter-helix loops on the extracellular side. In the previously reported apo structure of Kv3.1, the S5-PH loop of the pore domain, also known as the "turret" region[43], is directed toward the channel's narrow pore[42]. Also, there is an interaction between the turret and a flexible S1–S2 loop, and the turret and S3–S4 loop are directed against

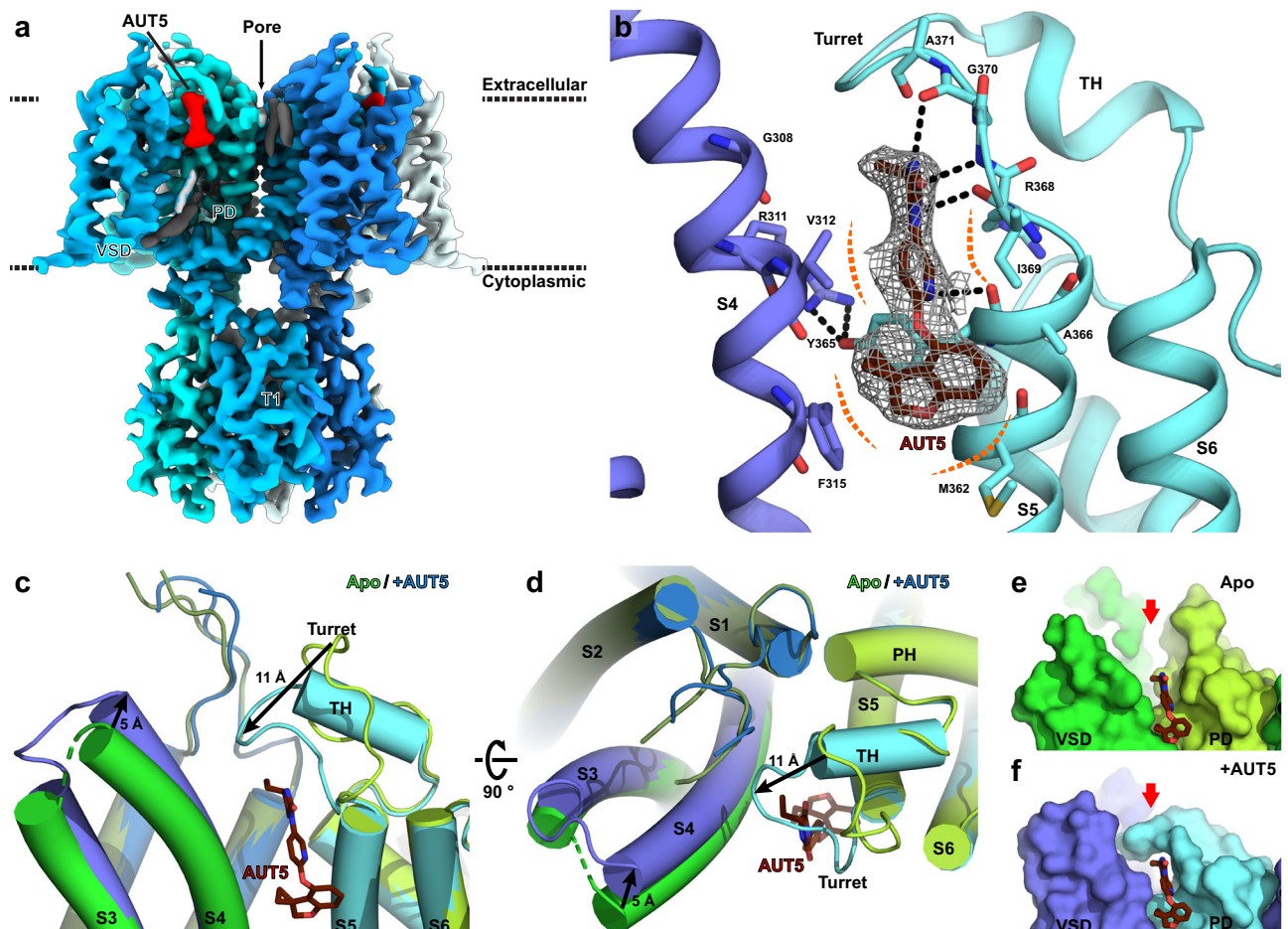

**Fig. 2 | High-resolution cryo-EM structure reveals the AUT5 binding site in Kv3.1. a** Overall cryo-EM reconstruction of Kv3.1/AUT5 complex. EM map feature for AUT5 is highlighted in red. **b** Cartoon representation of AUT5-binding site. Key residues for interaction with AUT5 (dark red) are highlighted as stick representations. Hydrogen bonds and van der Waals contacts are shown as black and orange dashed lines, respectively. The black dashed lines indicate potential polar interactions within a distance ≤3.2 Å. **c, d** Cartoon representations of conformational change of AUT5-binding site upon compound binding. Compared to Kv3.1 in apo state (green, PDB ID: 7PHI), Kv3.1 in AUT5-bound state (blue/cyan) have S3–S4 loop (blue) and Turret (cyan) shifted towards AUT5 by 5 Å and 11 Å, respectively. The C-terminal half of turret organizes into a two-turn alpha helix-like structure (TH). **e, f** Surface representations of AUT5-binding site of Kv3.1 in apo state (**e**) and AUT5-bound state (**f**). AUT5-binding pocket (red arrow) is open toward extracellular side in apo state, but it is closed by the turret in the modulator-bound state.

each other, effectively leaving the AUT5-binding cavity "open" on the extracellular side (Fig. 2c)[42]. In the AUT5-bound conformation, we observed remarkable rearrangements. The turret is shifted toward the VSD by 11 Å to close the cavity around the ID group of AUT5 compound for an induced fit, leading to interactions not only with AUT5 but also with S1–S2 and S3–S4 loops (Fig. 2d). This suggests that AUT5's ID group moiety is important for stabilizing interactions between S1–S2 loop and the turret, which we previously hypothesized to be a secondary site for conformational energy transfer from the VSD to the pore domain via electromechanical coupling[42]. Moreover, the C-terminal half of the turret (D375-E380) transitions from an unstructured loop in the apo state to a two-turn alpha helix in the AUT5-bound state, which would further stabilize this region in the ligand-bound state (Fig. 2b–d; Supplementary Information, Fig. S7). This observation has two potential consequences: (1) interactions between S1–S2, S3–S4 and the turret are critical for the transition between open and closed states in Kv3.1, and (2) positive modulation of Kv3.1 (and Kv3.2) by AUT5 is caused by the reorganization of these interactions (Fig. 2e, f).

AUT1 is an AUT5 analog and a positive modulator of Kv3.1 and Kv3.2, albeit its potency is lower than that of AUT5 (Supplementary Information, Fig. S3)[31,33]. AUT1's main structural difference is the substitution of the AUT5's hydrophobic SBC group with a 3-methoxy-4-methylphenyl (MMPh) group (Supplementary Information, Fig. S3).

Therefore, AUT1 would bind at the same site but have a different electron density map compared to AUT5, particularly in the area corresponding to the SBC/MMPh moiety. To test this hypothesis, we additionally solved the cryo-EM structure of the Kv3.1/AUT1 complex and found that the map feature corresponding to the MMPh moiety for AUT1 is both unambiguous and distinct from AUT5's SBC group (Supplementary Information, Fig. S7), leading us to conclude that the electron density maps of the Kv3.1/AUT5 and Kv3.1/AUT1 complexes resolved the expected differences and thereby enable correct modeling of the PAMs' binding modes.

To independently validate the AUT5 binding site, we also leveraged the previously solved apo structure of the Kv3.1 channel[42] and used blind docking calculations to determine the most likely location of the AUT5 binding site across the molecular surface of Kv3.1 and Kv3.2 channels (Methods). In the absence of any fingerprint of the compound on the channel structure, blind docking calculations consistently confirmed the location of the binding site for AUT5 and its configuration in the binding cavity of both Kv3.1 and Kv3.2 (Supplementary Results and Discussion, Figs. S11). In contrast, blind docking calculations with AUT5-insensitive Kv channel controls of known structure revealed that the AUT5 binding site found in Kv3.1 and Kv3.2 is not present in Kv1.2, Kv1.2-2.1 chimera and Kv4.2. (Supplementary Results and Discussion, Fig. S12). Over a total of 38 independent

docking solutions, focused docking calculations in which ligand interactions are resolved within a searching space spanning the binding site of the molecule additionally show that the bound configurations of AUT1 and AUT5 can be reproduced with best structural superpositions of 2.33 and 2.60 Å RMSD, respectively (Methods; Fig. 3). Also, by taking into consideration docking solutions that best describe the bound configurations of AUT1 and AUT5, we calculated the interaction energies of these compounds to determine whether binding site interactions could account for their differential potencies. In terms of the $\Delta V_{0.5}$ induced by these compounds, AUT5 is ~6 times more potent than AUT1 (Supplementary information, Fig. S3). Consistent with the greater modulatory potency of AUT5, the docking calculations revealed a broader and more energetically stable landscape of binding configurations of AUT5 over AUT1 (Fig. 3c, d). However, the interaction energies of the specific docking solutions that best reproduce the cryo-EM structures of the Kv3.1/AUT1 and Kv3.1/AUT5 complexes are very similar (−7.5 kcal/mol and −7.4 kcal/mol, respectively) (Fig. 3e, f). Therefore, differential concentration-dependent effects of the compounds cannot be entirely explained in terms of binding-site affinities. These calculations suggest that other

molecular factors associated with the transduction mechanism may additionally determine the differential functional potencies of these compounds.

## Functional mapping of the AUT5 and AUT1 binding determinants in Kv3.1

Guided by the high-resolution structural features of the identified binding site, we pursued functional validation of the residues that shape the cavity and established possible contacts with the bound compounds. Thus, we created a set of single Kv3.1 point mutations to change the shape and volume of the binding site at the bottom of the cavity, which accommodates the distinct SBC and MMPh groups of AUT5 and AUT1, respectively. This set included two mutations in the S4 helix (V312L and F315A), three mutations in the S5 helix (M362L, Y365A and A366L) and one mutation in the S6 helix (V416L). Although A366L and V416L do not appear to make direct contact with the compounds, we hypothesize that they may indirectly determine the cavity's shape (Fig. 2). We expressed these mutants in *Xenopus* oocytes and used TEVC to characterize the resulting K$^+$ currents in the absence and presence of AUT5 or AUT1

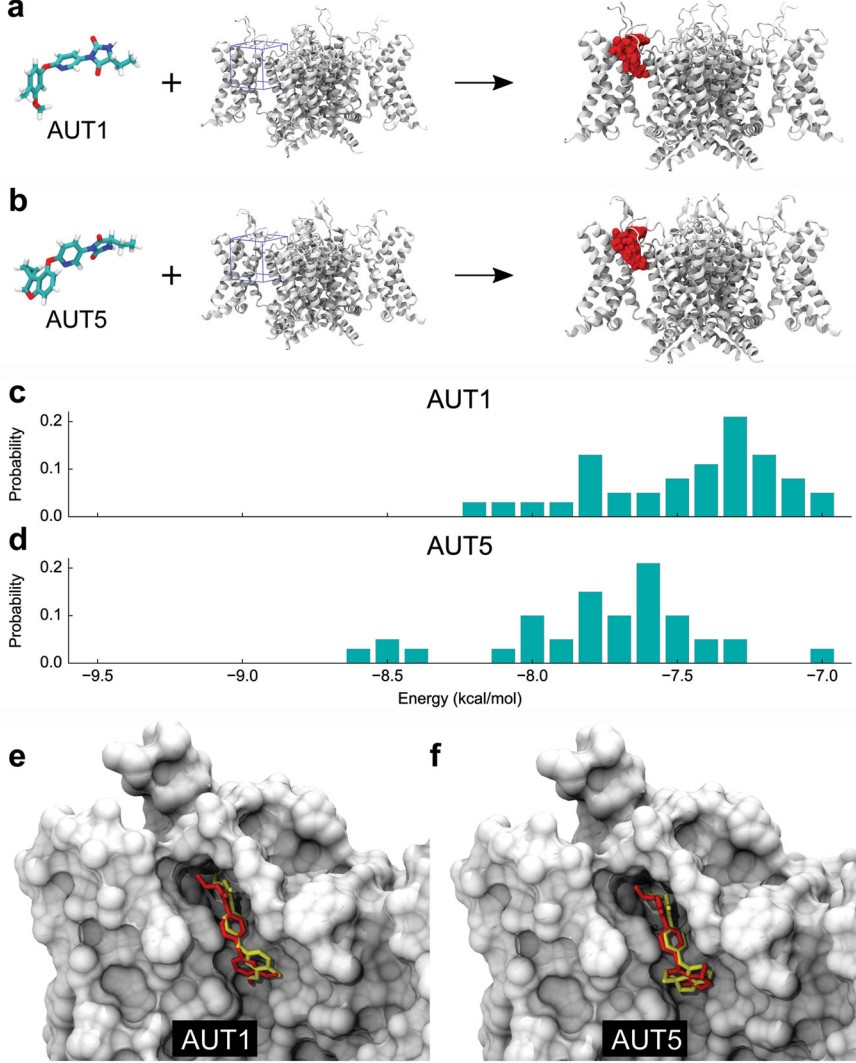

**Fig. 3 | Comparative analysis of AUT1 and AUT5 docking solutions.**
**a**, **b** Molecular structures of AUT1 and AUT5, Kv3.1 (white) and docking solutions (red) across the search grid volume (blue). **c**, **d** Distribution of binding energy values in docking. Each distribution contains 38 independent solutions. **e**, **f** Shown are docking solutions (red) that best reproduce the cryo-EM determined

configurations (yellow) of AUT1 and AUT5, with structural superposition of 2.33 and 2.60 Å RMSD and interactions energies of −7.5 and −7.4 kcal/mol, respectively. The negative shift between energy distributions in (**c**, **d**) suggests that the overall set of bound configurations of AUT5 is modestly more stable than that of AUT1.

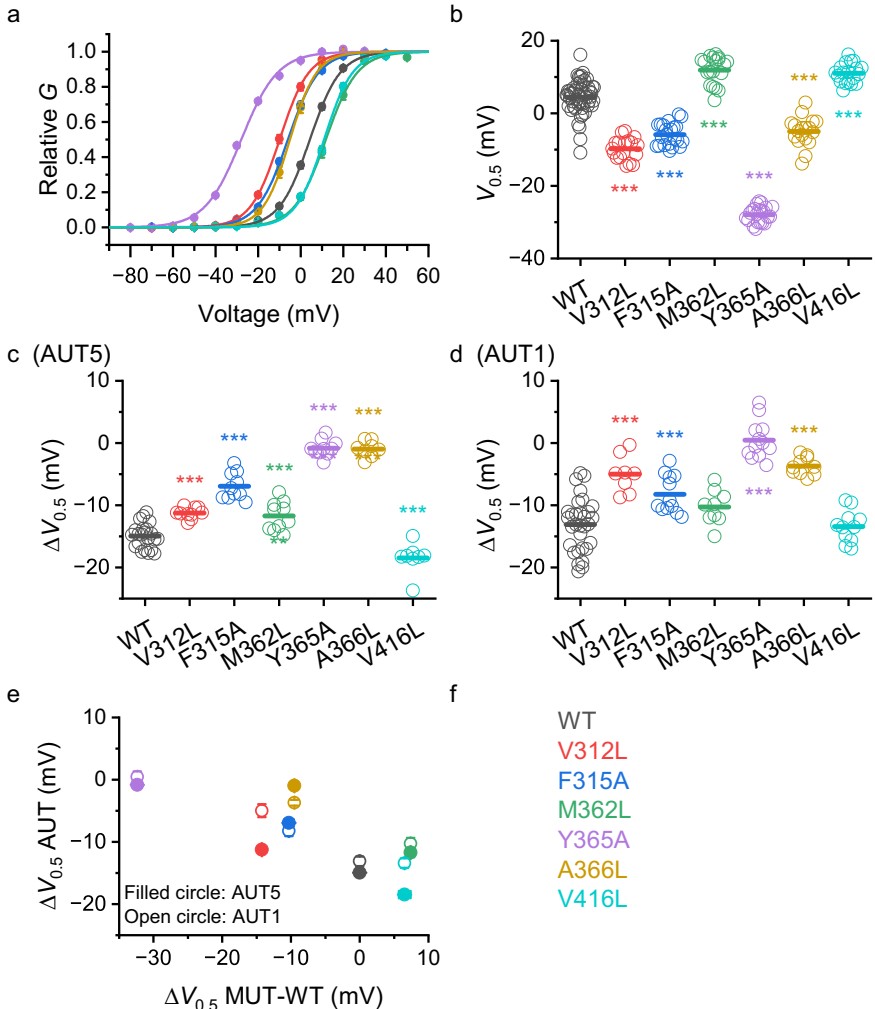

**Fig. 4 | Kv3.1 mutations in the binding site for AUT5 and AUT1 alter voltage dependence and modulation by the compounds. a** $G_p$ – $V_c$ relations of Kv3.1 wild-type (WT) and the Kv3.1 mutants listed, and color-coded on panel f. In most cases, the SEM bars are obscured by the symbols. The solid lines are the best fit of the Boltzmann function (Methods). Scatter plots comparing the $V_{0.5}$ values of Kv3.1 wild-type and the indicated mutants (**b**) alone, (**c**) with AUT5, and (**d**) with AUT1. $V_{0.5}$ is derived from the best-fit Boltzmann function. **c, d** Scatter plots comparing the $\Delta V_{0.5}$ induced by AUT5 and AUT1, respectively. One-way ANOVA was used to evaluate changes relative to WT. *** <0.001. **e** Correlation plot of the mean $\Delta V_{0.5}$ induced by AUT5 or AUT1 vs. the mean $\Delta V_{0.5}$ caused by the mutations from panels

(**b–d**). The SEM bars are in most cases smaller than the symbols. **f** Color coded names of the Kv3.1 mutations characterized on this figure. Number of oocytes tested with AUT5 (**c, e**): 22 (WT), 10 (V312L), 10 (F315A), 10 (M362L), 10 (Y365A), 8 (A366L), and 8 (V416L). Number of oocytes tested with AUT1 (**d, e**): 31 (WT), 8 (V312L), 12 (F315A), 9 (M362L), 12 (Y365A), 10 (A366L), and 11 (V416L). **a** and **b** are the summary of the above experiments before AUT5 or AUT1 was applied. **b** $P$ = 4.7E-19 (V312L), 9.4E-15(F315A), 2.4E-8(M362L), 3.0E-44(Y365A), 3.9E-11(A366L), 1.5E-7(V416L). **c** $P$ = 3.0E-6(V312L), 1.5E-11(F315A), 2.6E-4(M362L), 2.2E-19(Y365A), 1.8E-17(A366L), 3.1E-4(V416L). **d** $P$ = 1.1E-5(V312L), 7.7E-4(F315A), 1.2E-12(Y365A), 3.8E-8(A366L).

(Methods; Fig. 4; Table S4). Consistent with the importance of the interface between the S4, S5 and S6 helices in voltage-dependent gating[44,45], the mutations induced hyperpolarizing $\Delta V_{0.5}$ shifts ranked as follows (mV): −32.4 (Y365A), −14.2 (V312L), −10.3 (F315A), −9.5 (A366L) (Figs. 4b and 4e); and the remaining mutations induced depolarizing shifts ranked as follows (mV): 7.4 (M362) and 6.5 (V416L) (Figs. 4b, e). It is notable that the hyperpolarizing mutations qualitatively mimic the positive modulation induced by AUT5 and AUT1. In addition, these substitutions inhibited the hyperpolarizing effect of AUT5 and AUT1 on the $G_p$-$V_c$ curve in a similar fashion (Fig. 4c, d). To some degree, the magnitude of the decreased positive modulation is inversely related to the magnitude of the hyperpolarizing shift already induced by the mutations under basal conditions (Fig. 4c–e). For instance, Y365A induced the greatest hyperpolarizing shift while also nearly abolishing positive modulation by AUT5 and turning the positive modulation by AUT1 into a slight negative modulation (depolarizing shift). However, A366L also nearly abolished the positive modulation by AUT5 and

inhibited the positive modulation by AUT1 to a lesser degree, despite inducing a modest hyperpolarizing shift under basal conditions. Similarly, V312A induced a modest hyperpolarizing shift under basal conditions and greatly reduced the positive modulation by AUT1, while inhibiting the positive modulation by AUT5 to a lesser degree. The least impactful Kv3.1 mutations were M362L and V416L. Whereas the M362L similarly inhibited the positive modulation by AUT5 and AUT1, V416L did not inhibit the positive modulation by AUT1, and slightly increased the positive modulation by AUT5. Overall, these results are consistent with the contributions of the mutated residues to the identified binding site in Kv3.1 and the efficacy of the modulation. However, reflecting the focused docking results (Fig. 3), the mutation-induced changes on the modulation by AUT5 or AUT1 are overall similar (Fig. 4e). Also, the identified contact sites in the binding cavity are highly conserved in Kv3.4, which is relatively less sensitive to positive modulation by AUT5 (Supplementary Information, Figs. S1, S4 and S9). Therefore, the selectivity of the modulation and the transduction mechanism

must be governed by other regions involved in allosteric gating control.

### The unique turret region of Kv3 channels determines the positive modulation by AUT5

We hypothesized that the structural basis of the highly selective positive modulation of Kv3.1 and Kv3.2 by AUT5 may concern discrete and specific structural differences outside the binding site. Rather than being involved in the binding of AUT5 and AUT1, these differences could also be responsible for the transduction mechanism that governs the positive modulation. Thus, we compared the sequences of Kv3 channels that exhibit robust positive modulation by AUT5 (Kv3.1 and Kv3.2) against those that lack this property (Kv3.4 and K-Shaw2) and found potentially significant differences in the cytoplasmic T1 domain, the extracellular S1-S2 loop and the extracellular turret region (Supplementary Information, Figs. S9 and S10). To test the potential contributions of these regions, we created the following constructs: deletion of the T1 domain (ΔT1-Kv3.2), replacing the S1-S2 loop of Kv3.2 with that of Kv3.4 (3.4×3.2/S1S2), and deletion of the turret (ΔTurret-Kv3.2) (Methods; Supplementary Information, Table S2). Under basal conditions, ΔT1-Kv3.2, 3.4×3.2/S1S2 and ΔTurret-Kv3.2 caused modest rightward shifts of the $G_p$-$V_c$ relation (<15 mV; Supplementary Information, Figs. S14). However, whereas ΔT1-Kv3.2, retained intact positive modulation by 2 µM AUT5, the 3.4 × 3.2/S1S2

chimera exhibited modestly decreased modulation (32% less than wild type Kv3.2) and the ΔTurret-Kv3.2 exhibited no modulation (Fig. 5). These results suggest that the S1-S2 loop may play a limited role in the positive modulation by AUT5; however, the presence of the turret is most critical. This is remarkable because Kv3.1 and Kv3.2 have nearly identical turret regions (Fig. 6; Supplementary Information, Fig. S9), and Kv channels that lack a homologous turret sequence, such as Kv1.2, Kv2.1, Kv4.2 and the *Drosophila* Kv3 ortholog K-Shaw2 (Fig. 6; Supplementary Information, Fig. S9), exhibit no modulation by 2 µM AUT5 (Supplementary Information, Fig. S1). Also, compared to Kv3.1 and Kv3.2, Kv3.4 has a turret with eight potentially significant differences in its amino acid sequence (Fig. 6). To determine whether the presence of a specific turret sequence can explain the differential modulation of Kv3.2 and Kv3.4 by AUT5, we created a Kv3.2 chimera in which we replaced its turret sequence with that of Kv3.4 ("Methods"; 3.4 × 3.2/Turret; Table S2) and characterized the $G_p$-$V_c$ relation of this chimera before and after bath application of 2 µM AUT5. Consistent with a contribution of the turret to gating under basal conditions, the 3.4 × 3.2/Turret exchange caused a 11.8 mV depolarizing shift of the $V_{0.5}$ ($P = 1.2 × 10^{-7}$; Supplementary Information, Fig. S15). More significantly, the 3.4 × 3.2/Turret exchange nearly eliminated the modulation by 2 µM AUT5, closely recapitulating the effect of the Kv3.2 turret deletion (Fig. 5), which strongly supports the role of the Kv3.2 turret as a critical determinant of the positive modulation by AUT5.

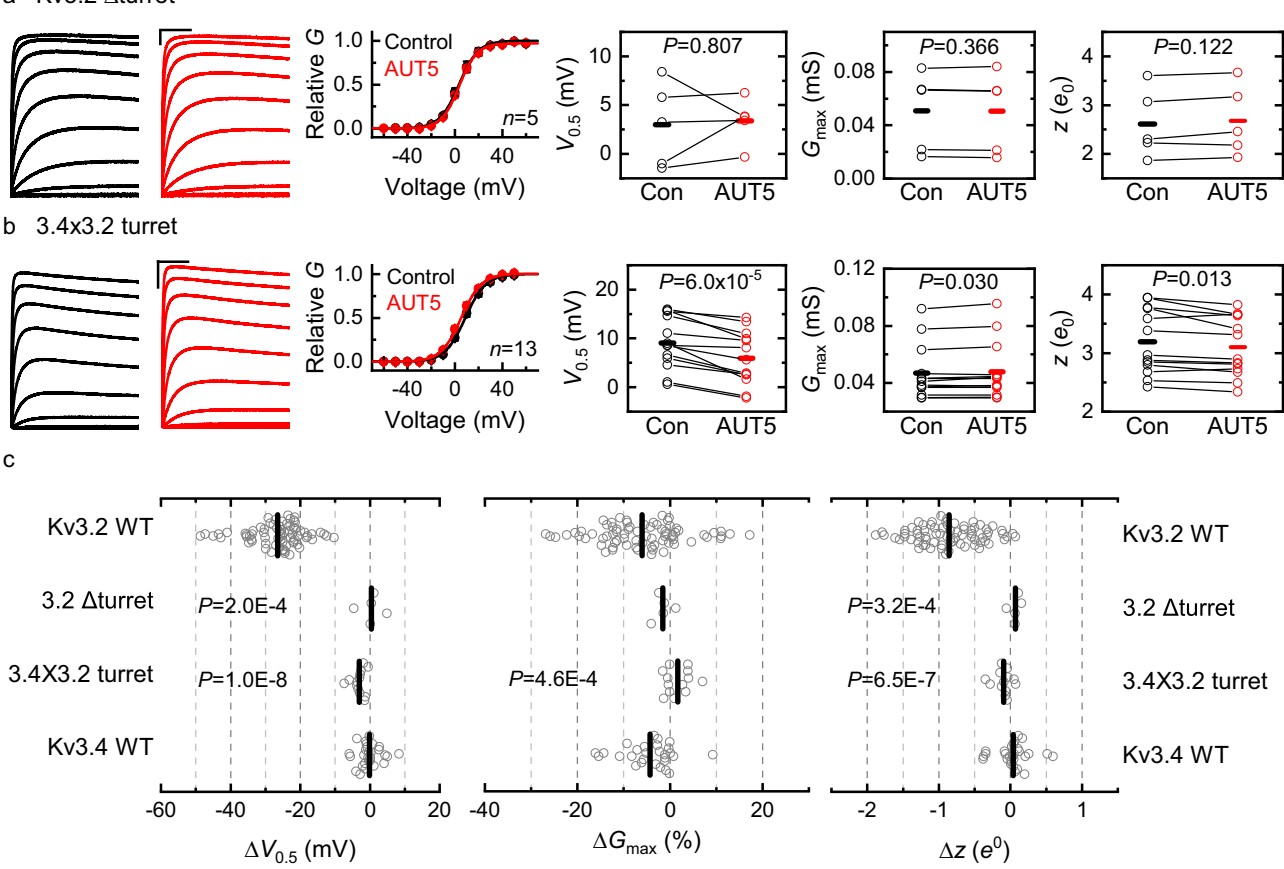

**Fig. 5 | The turret region is necessary for the positive modulation of Kv3.2 by AUT5. a, b** Representative families of Kv3.2ΔTurret and 3.4 × 3.2 Turret currents (left) before (black) and after (red) bath application of 2 µM AUT5, and the aggregate of $G_p$-$V_c$ curves with their corresponding analysis of $V_{0.5}$, $G_{max}$ and z (right). SEM bars are obscured by the symbols. The $G_p$-$V_c$ curves are normalized relative to the control in the absence of a compound. Representative currents were evoked by the voltage protocol shown on Fig. S1d and described in the corresponding legend, and the solid lines across the symbols of the $G_p$-$V_c$ curves are the

best fit to the 1st-order Boltzmann equation (Methods). Scale bars represent 1 µA and 100 ms. **c** Aggregate scatter graphs of the AUT5-induced changes in $V_{0.5}$, $G_{max}$ and z from individual oocytes. Short vertical bars indicate the mean values. The sample sizes of the wild-type groups are as indicated in Fig. S1. For each mutant, the indicated P values evaluate differences relative to wild-type Kv3.2 (Kruskal-Wallis test). The results from Kv3.2ΔTurret relative to wild-type Kv3.4 are indistinguishable. Each symbol represents a measurement from a single oocyte. Number of oocytes: 5 (ΔTurret), 13 (3.4 × 3.2 Turret), 74 (Kv3.2 WT), and 28 (Kv3.4 WT).

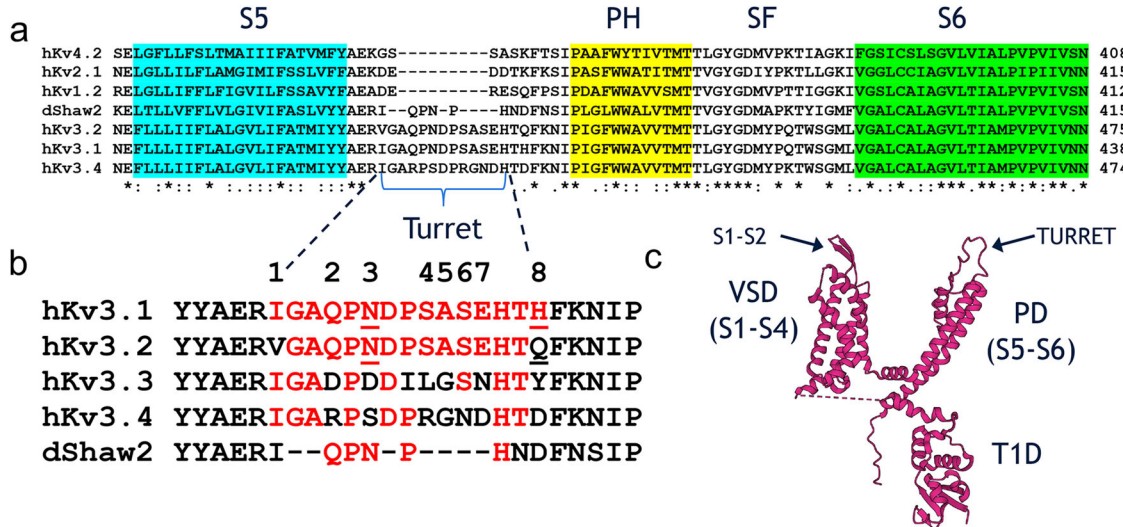

**Fig. 6 | Kv3 channels with differential AUT sensitivity have distinct turret sequences. a** Sequence alignment comparing transmembrane segment S5, turret loop, pore helix (PH), selectivity filter (SF) and transmembrane segment S6 from homologous Kv channels. Note that the turrets of hKv4.2, hKv2.1, hKv1.2 and dShaw2 are shorter compared to those of hKv3.1, hKv3.2 and hKv3.4. **b** Detailed comparison of amino acid sequences in the turrets of Kv3 channels (dShaw is a *Drosophila* homolog). Note that, except for the flanking residues at the numbered positions 1 and 8, Kv3.1 and Kv3.2 have identical turret regions. **c** Topology of a single Kv3.1 subunit depicting the major domains: cytoplasmic T1 domain (T1D), voltage-sensing domain (VSD = S1–S4) and pore domain (PD = S5–S6). Arrows indicate the extracellular S1–S2 loop and turret region.

To conclusively demonstrate that the turret is necessary and sufficient to determine the positive modulation by AUT5, we must also show that the presence of the Kv3.2 turret in Kv3.4 is conversely sufficient to confer a significant AUT5-induced hyperpolarizing shift of the $G_p$-$V_c$ relation. Thus, we created a Kv3.4 chimera in which we replaced its turret sequence with that of Kv3.2 (Methods; 3.2 × 3.4/Turret; Supplementary Information, Table S2) and, as above, characterized the $G_p$-$V_c$ relation before and after bath application of 2 µM AUT5 (Fig. 7). The $G_p$-$V_c$ relation of the 3.2 × 3.4/Turret chimera under basal conditions was leftward shifted relative to that of the wild-type Kv3.4 (−7.8 mV; Supplementary Information; Fig. S16), consistent again with a contribution of the turret to voltage-dependent gating. Further supporting this contribution, the deletion of the turret in Kv3.4 produced a similar leftward shift (−11 mV; Supplementary Information; Fig. S16). Upon bath application of 2 µM AUT5 to oocytes expressing the 3.2 × 3.4/Turret chimera, the conductance starts to increase slightly at more negative voltages relative to the control indicating a hint of positive modulation (Fig. 7a). Also, the $G_p$-$V_c$ relation in the presence of AUT5 crosses the control $G_p$-$V_c$ relation and displays a modest decrease of the $G_{max}$ (Fig. 7a). Considering these effects, we hypothesized that the presence of Kv3.4 fast inactivation truncated the conductance at the most depolarized voltages, which partially obscured the expected AUT5-induced hyperpolarizing shift of the $G_p$-$V_c$ relation.

Kv3.4 undergoes fast inactivation and is relatively insensitive to AUT5 modulation (Supplementary Information, Figs. S1, S4). By contrast, Kv3.1 and Kv3.2 undergo little to no inactivation and exhibit robust positive modulation by AUT5 (Supplementary Information, Fig. S1). Therefore, we conducted experiments to determine how fast Kv3.4 inactivation may have obscured the positive modulation by AUT5 in the 3.2 × 3.4/Turret chimera. Accordingly, we leveraged previous studies that demonstrated quick elimination of Kv3.4 fast inactivation by protein kinase C (PKC) – dependent phosphorylation of the channel's N-terminal inactivation domain[46]. Bath application of a phorbol ester to activate PKC (50 nM phorbol 12-myristate 13-acetate, PMA) in oocytes expressing Kv3.4 eliminated the channel's fast inactivation over a period of ~10 min (Supplementary Information, Fig. S17). This modulation, however, did not significantly change the low sensitivity of the wild-type Kv3.4 to modulation by 2 µM AUT5, demonstrating that fast inactivation per se does not mask positive modulation of the wild-type Kv3.4 by AUT5 (Supplementary Information, Fig. S17). We next exposed oocytes expressing the 3.2 × 3.4/Turret chimera to 50 nM PMA and, once fast inactivation was eliminated, we characterized the $G_p$-$V_c$ relations before and after bath application of 2 µM AUT5, while keeping PMA constant in the bath (Fig. 7b). Consistent with the hypothesis that fast inactivation partially obscured the expected positive modulation of the 3.2 × 3.4/Turret chimera by AUT5 after PMA treatment, the 3.2 × 3.4/Turret chimera displayed a significant AUT5-induced hyperpolarizing shift of the $G_p$-$V_c$ relation ($\Delta V_{0.5} = -15.7 \pm 1.0$ mV; Fig. 7), which phenocopies ~60% of the positive modulation observed with wild-type Kv3.2 ($\Delta V_{0.5} = -26.5 \pm 0.9$ mV; Figs. 1, 7b).

## Discrete differences in the turret region dictate the highly selective positive modulation of Kv3 channels by AUT5

There are eight differences (numbered 1–8) between the putative turret sequences of Kv3.2 and Kv3.4 (Fig. 6). According to the cryo-EM structure of human Kv3.1[42], however, four differences are located within the segment of the extracellular loop that shapes the turret, including N3, S4, A5 and S6 in Kv3.1 and Kv3.2, and the corresponding S3, R4, G5 and N6 of Kv3.4 (Fig. 6). The SAS triad is especially interesting because it is close to the extracellular S1-S2 loop of the VSD at the inter-subunit interface in the cryo-EM structure of Kv3.1 (Fig. 2, Fig. S10). Therefore, to determine whether specific differences within the turret are responsible for the differential sensitivities of these Kv3 channels to modulation by AUT5, we replaced individual Kv3.4 residues for those at the equivalent positions in Kv3.2 (N3S, S4R, A5G and S6N) and characterized the $G_p$-$V_c$ relations before and after exposing the oocytes expressing these mutants to 2 µM AUT5. Consistent with the results from the Kv3.2 turret deletion and turret chimera under basal conditions, single Kv3.2 turret mutations caused slight depolarizing shifts of the $G_p$-$V_c$ relation relative to the voltage dependence of wild-type Kv3.2 (Supplementary Information, Fig. S15). More significantly, however, these mutations had significant effects on the modulation of voltage-dependent gating by AUT5 (Fig. 8a; Supplementary Information, Fig. S18). Compared to wild-type Kv3.2, both N3S

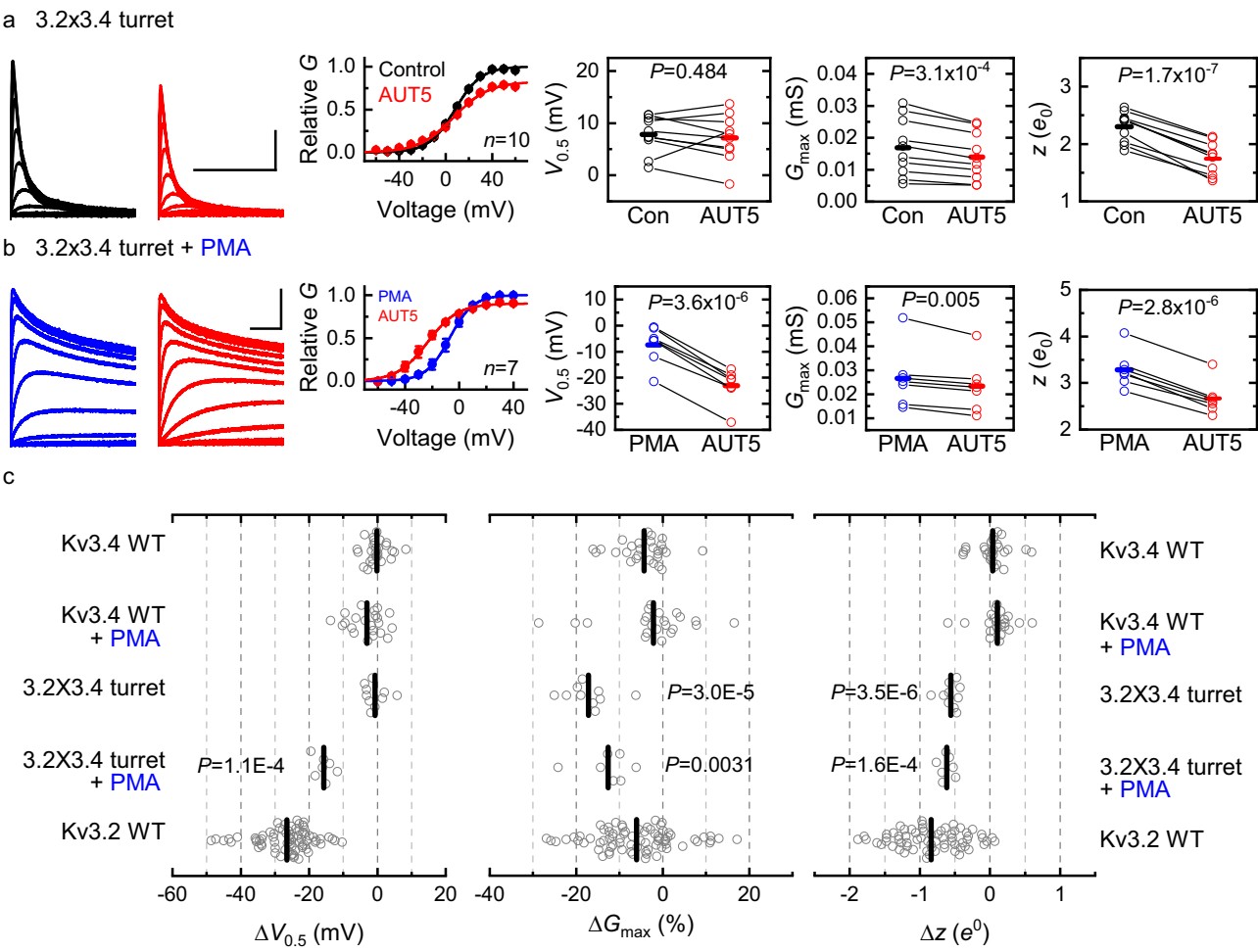

**Fig. 7 | The Kv3.2 turret region confers positive modulation by AUT5 in Kv3.4 upon elimination of fast inactivation. a, b** Representative families of 3.2 × 3.4Turret currents (control and post-PMA, left, top and bottom, respectively) before (black) and after (red) bath application of 2 µM AUT5 and the aggregate of $G_p$-$V_c$ curves with their corresponding analysis of $V_{0.5}$, $G_{max}$ and z (right). SEM bars are obscured by the symbols. The $G_p$-$V_c$ curves are normalized relative to the control in the absence of a compound. Representative currents were evoked by the voltage protocol shown in Fig. S1d, and the solid lines across the symbols of the $G_p$-$V_c$ curves are the best fits of the 1st-order Boltzmann equation ("Methods"). Blue traces and symbols are from oocytes exposed to 50 nM PMA before applying AUT5.

PMA remained in the chamber until the end of the experiment. **c** Aggregate scatter graphs of the AUT5-induced changes in $V_{0.5}$, $G_{max}$ and z from individual oocytes. Short vertical bars indicate the mean values. The Kv3.2 WT results are replotted here as a reference. The sample sizes of the wild-type groups are as indicated in Fig. S1. For each mutant, the indicated P values evaluate differences relative to wild-type Kv3.4 either in the presence or absence of PMA (Kruskal-Wallis test). Each symbol represents a measurement from a single oocyte. Number of oocytes: 7 (3.2 × 3.4 Turret), 10 (3.2 × 3.4 Turret + PMA), 28 (Kv3.4 WT), 22 (Kv3.4 WT + PMA), and 74 (Kv3.2 WT).

and A5G inhibited the AUT5-induced hyperpolarizing shift, albeit the neutralizing effect of the latter was modestly greater ($\Delta V_{0.5} = -14.2 \pm 0.6$ mV and $-11.3 \pm 1.0$ mV, respectively) (Fig. 8a). By contrast, S4R increased the $\Delta V_{0.5}$ ($-42.2 \pm 2.7$ mV) and S6N had no effect ($-25.9 \pm 1.2$ mV) (Fig. 8a). Since no individual mutation completely neutralized the positive modulation by AUT5 and the effects were qualitatively diverse (decrease, increase and no effect), we explored the effects of combined mutations. Thus, we created three Kv3.2 mutant constructs that combined the mutation that had the weakest neutralizing effect (N3S) with substitutions at each position of the SAS triad (N3S/S4R, N3S/A5G, N3S/S6N), and a triple mutation that exchanged the RGN triad of Kv3.4 for the SAS triad of Kv3.2 (S4R/A5G/S6N). These mutations under basal conditions also caused slight-modest depolarizing shifts of the $G_p$-$V_c$ relations relative to the voltage dependence of wild-type Kv3.2 (Supplementary Information, Fig. S15). In response to AUT5 application, double mutations similarly neutralized the AUT5-induced hyperpolarizing shift, albeit the neutralization of the positive modulation was partial (Fig. 8a; Supplementary Information, Fig. S18). The triple triad mutation, in

contrast, was more effective at neutralizing the AUT5-induced hyperpolarizing shift (Fig. 8a; Supplementary Information, Fig. S18), an effect that nearly matched the complete neutralization produced by a complete exchange of the Kv3.4 turret for the Kv3.2 turret (Fig. 8a). These results demonstrate that discrete Kv3.2 turret substitutions that exchange Kv3.4 residues for Kv3.2 residues (single, double, and triple) are sufficient to neutralize the positive modulation by AUT5, albeit these effects are generally partial and do not suggest simple additivity. Notably, however, complete turret conversion from Kv3.2 to Kv3.4 (equivalent to eight substitutions, V1I/Q2R/N3S/S4R/A5G/S6N/E7D/ Q8D) and turret deletion eliminate the positive modulation of Kv3.2 by AUT5 (Fig. 8a), making the low AUT5 sensitivity of these constructs indistinguishable from that of wild type Kv3.4.

If discrete Kv3.2 turret substitutions affecting critical determinants can neutralize the positive modulation by AUT5, discrete reciprocal substitutions in Kv3.4 should confer positive modulation by AUT5. To test this hypothesis, we created the following Kv3.4 turret mutations (numbered as explained above): S3N, R4S, G5A, N6S and S3N/N6S (Fig. 6). We then assessed their function and modulation by

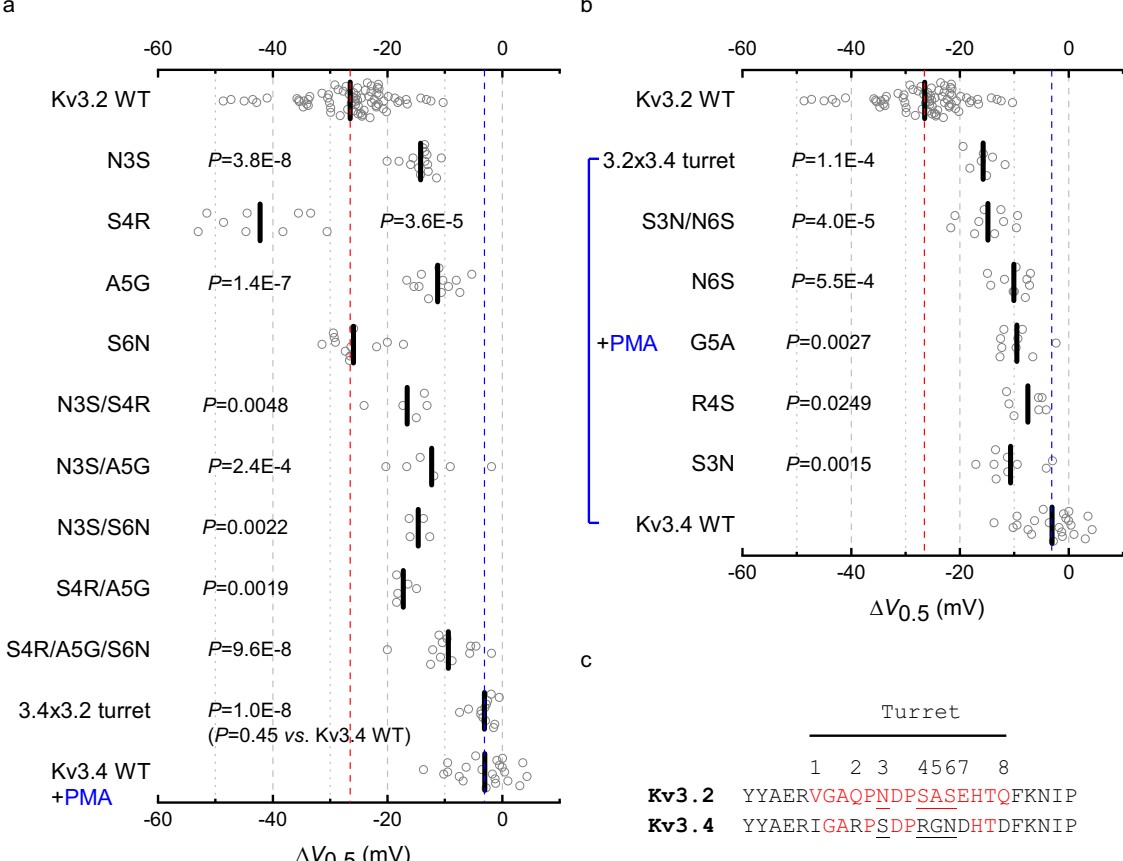

**Fig. 8 | Discrete turret mutations can eliminate or confer positive modulation by AUT5 in Kv3.2 and Kv3.4, respectively. a, b** Scatter plot of $V_{0.5}$ changes induced by 2 μM AUT5. The red and blue dashed lines are shown as the reference values of the $V_{0.5}$ of Kv3.2 WT and Kv3.4 WT, respectively. The location of the mutations is according to the key shown in (**c**), Fig. 6 and Table S2, which indicate eight turret differences between Kv3.2 and Kv3.4. Modulation of Kv3.4 WT and Kv3.4 mutants by 2 μM AUT5 was tested in the presence of 50 nM PMA (blue bracket) to eliminate fast inactivation as shown in Fig. 7. Short vertical bars indicate the mean values. The indicated $P$ values evaluate differences relative to wild-type Kv3.2 in (**a**) and wild-type Kv3.4 in the presence of PMA in (**b**) (Kruskal-Wallis test). The sample sizes (number of oocytes) in (**a**) are: N3S, $n = 15$; S4R, $n = 9$; A5G, $n = 12$; S6N, $n = 12$; N3S/A5G, $n = 6$; N3S/S4R, $n = 5$; N3S/S6N, $n = 4$; S4R/A5G, $n = 5$; S4R/A5G/S6N, $n = 12$. The sample sizes (number of oocytes) in (**b**) are: S3N, $n = 9$; R4S, $n = 7$; G5A, $n = 9$; N6S, $n = 10$; S3N/N6S, $n = 10$. Each symbol represents a measurement from a single oocyte. **c** Sequence alignment comparing turret loop of Kv3.2 and Kv3.4.

AUT5 following exposure to PMA as described above (Supplementary Information; Fig. S17). In the presence of PMA, these mutants displayed voltage dependence that was slightly hyperpolarized compared to the voltage dependence of wild-type Kv3.4 (Supplementary Information; Fig. S16). Supporting the important role of specific Kv3.2 turret residues, all mutations conferred similar AUT5-induced positive modulation that is greater than what is observed with wild-type Kv3.4 in the presence of PMA (Fig. 8b; Supplementary Information, Fig. S19). Most significantly, the double Kv3.4 mutation S3N/N6S was sufficient to match the AUT5-induced positive modulation of the Kv3.4 chimera with a turret fully converted to that of Kv3.2 (Fig. 8b). Overall, the chimeras and discrete reciprocal substitutions between Kv3.2 and Kv3.4 strongly suggests that discrete structural differences between the turret regions of Kv3 channels account for the highly specific positive modulation of Kv3.1 and Kv3.2 channels by AUT5.

## Discussion

In this study, we employed an orthogonal approach that included cryo-EM, extensive site-directed mutagenesis, voltage-clamping and docking calculations to determine the mechanism of action of a distinct class of highly selective positive allosteric modulators of Kv3 channels with structural precision. As a result, we solved two distinct ligand-bound structures of the Kv3.1 channel at near-atomic resolution, which revealed a binding site for small-molecule modulators and demonstrated gating interactions that underlie the transduction mechanism. Furthermore, the mutational, functional, and computational analyses revealed the structural basis of modulation selectivity, and helped elucidate a transduction mechanism that ultimately dictates the functional potency of the compounds.

Although Kv3.2 exhibits the greatest positive modulation by AUT5, Kv3.1 also exhibits substantial positive modulation by this compound, in contrast to the relatively low sensitivities of closely and more distantly related Kv channels, which also have a domain-swapped architecture (Supplementary Information, Fig. S1). Therefore, leveraging on our previous cryo-EM work with Kv3.1, we solved the high-resolution cryo-EM structure of the Kv3.1/AUT5 complex and corroborated the location and structural features of the binding site by solving the structure of Kv3.1 bound to AUT1, another imidazolidine-dione derivative that is less potent than AUT5 (Supplementary Information, Fig. S3, Table S4). We found four equivalent AUT5 binding sites facing the extracellular side of the four-fold symmetric Kv3.1 assembly, strategically located in the inter-subunit interface sites mainly lined by apolar amino acid side chains from the membrane-spanning segments S4 (from the VSD), and S5 (from the PD of the neighboring subunit). Also, a comparison of the apo and bound structures allowed visualization of the mechanism of action. Demonstrating an induced fit upon AUT1 or AUT5 binding, the extracellular turret region undergoes rearrangements of its secondary and tertiary structures to trap the

compound in the binding cavity and establish hydrogen bond interactions between the turret's backbone and the polar moiety of the compound. This modulator binding site is entirely different from that reported for other modulators of voltage-gated ion channels in nearby regions, such as LuAG00563, ztz240, ICA13431, GX-936 and ProTx2[47–51] (Supplementary Information, Fig. S7j). LuAG00563 is especially relevant for comparison because it represents another class of positive modulators of Kv3 channels. The recently reported cryo-EM structure of the Kv3.1/LuAG00563 complex showed the ligand occupying an inter-subunit binding site that faces the intracellular side of the Kv3.1 assembly. This site is lined by segments S1 and S4 from one subunit and segment S5 from a neighboring subunit, with contacts mainly involving apolar side chains that are not overlapping with contact sites involved in AUT5 binding (Supplementary Information, Fig. S7j)[51]. Moreover, the Kv3.1/LuAG00563 complex did not reveal multiple equivalent binding sites and conformational changes when compared to the Kv3.1 apo structure. Therefore, imidazolidinedione compounds and LuAG00563 are positive modulators of Kv3 channels with distinct binding sites and mechanisms of action. The high selectivity of AUT5 for Kv3.1 and Kv3.2 may additionally set the imidazolidinedione derivatives further apart from LuAG00563 and other ion channel modulators.

In the binding site of Kv3.1, AUT5 and AUT1 make contacts with the side chains from V312 and F315 in the S4 helix, and M362, Y365, I369 and A371 in the S5 helix, which are highly conserved in Kv3.1, Kv3.2 and Kv3.4 (Supplementary Information, Fig. S9). This is intriguing because, compared to Kv3.1 and Kv3.2, Kv3.4 expressed in *Xenopus* oocytes exhibits little to no positive modulation by AUT5 (Supplementary Materials; Fig. S1). Also, Kv3.4 stably expressed in HEK293 cells displays a slight but not significant positive modulation at 2 μM AUT5, while at 10 μM AUT5 a weak positive modulation is seen with small depolarizations and negative modulation is observed with strong depolarizations (Supplementary Materials; Fig. S4). It is also intriguing that, functionally, AUT5 is significantly more potent than AUT1, and yet we found that the binding configurations, contacts sites and binding energies of these compounds are nearly identical in Kv3.1 (Fig. 3). The latter was further validated by independently demonstrating that binding site point mutations of Kv3.1 similarly inhibit positive modulation by AUT1 and AUT5 (Fig. 4). It is, therefore, more likely that intrinsic interactions outside the binding pocket dictate selectivity, functional potency, and the transduction mechanism responsible for the positive modulation.

Overall, our results conclusively demonstrate that the high selectivity of AUT5 for Kv3.1 and Kv3.2 is mainly determined by their extracellular turret region. Considering the appearance of the extracellular S5-PH loop in the crystal structure of the bacterial KcsA channel, Doyle et al. first used the term turret to describe this region[43]. Subsequent studies discovered that the turret is highly variable among eukaryotic Kv channel structures, where, in addition to shaping the selectivity filter, it (1) helps determine binding of inhibitory toxins, nanobodies and synthetic compounds[52–55]; (2) contributes to voltage-dependent gating rearrangements[56,57]; (3) interacts with an ancillary protein to modulate binding of polyunsaturated fatty acids[58]; and (4) is a structural determinant of C-type inactivation[59,60]. Expanding the importance of the turret, our work demonstrates the allosteric role of the turret in the mechanism of action of highly selective small molecule modulators of Kv channels, such as AUT1 and AUT5. The results showed that a reciprocal exchange of the turret sequences between Kv3.2 and Kv3.4 eliminates and confers positive modulation by AUT5. Moreover, discrete mutations identified a triad of consecutive residues (SAS in Kv3.2 and RGN in Kv3.4) as the most critical determinants of the selective positive modulation or lack thereof. The apparent lack of additive effects of the mutations is more consistent with turret rearrangements induced by AUT5 than with direct contributions of the mutated residues to contact sites (Fig. 8). This was directly corroborated by the high-resolution cryo-EM structures of the Kv3.1/AUT5 and Kv3.1/AUT1 complexes, where it is evident that the turret undergoes major rearrangements of its secondary and tertiary structures to trap the compound in the binding site and establish interactions that are responsible for the transduction mechanism that underlies the positive modulation.

Canonical and non-canonical mechanisms of gating in voltage-dependent ion channels implicate multiple interactions between the VSD and PD[61]. It is, therefore, not surprising that binding site mutations that affect the VSD-PD interface between neighboring subunits under basal conditions had effects on voltage dependence that generally emulate the positive modulation induced by AUT5 and AUT1 (Fig. 4a, b). It is, however, especially interesting that these mutations also neutralize the positive modulation (Fig. 4c–e), suggesting that they have a negative impact on the striking conformational changes induced by the compounds, especially in the turret region. It has been previously suggested that the turret in Kv1.2 channels undergoes opening rearrangements that may change its relationship with the S1–S2 loop of the VSD[56]. The turret may thereby indirectly affect the conformation of the S3–S4 loop and, as a result, influence voltage-dependent gating. There is also ample evidence demonstrating that the S3–S4 loop of Kv channels undergoes gating rearrangements resulting from the voltage-dependent movements of the S4 voltage sensor that governs gating[62–64]. Consistent with these findings, our results strongly suggest that the structural underpinnings of the positive modulation by AUT5 involve turret rearrangements induced by compound binding in the identified cavity and reorganization of the interactions between the turret and the VSD.

From the apo and bound structures of Kv3.1 and insights from MD simulations in the apo conformation, we have identified a mechanism of action that is generally consistent with all observations. We propose that upon entering the identified cavity and establishing stabilizing hydrophobic interactions, AUT5 (or AUT1) triggers major rearrangements of the turret structure (Movie #1). Specifically, the turret undergoes two major changes upon the formation of hydrogen bonds between its N-terminal region and the compound's polar ID moiety. First, the turret's C-terminal region, including the SAS triad in Kv3.1 and Kv3.2, transitions from an unstructured to a two-turn helix-like structure. Second, the rest of the turret moves 11 Å toward the S3–S4 loop of the VSD and folds over to cap the compound in the binding site, via an induced-fit mechanism. In the apo conformation, the turret is near a relatively inflexible S1–S2 loop (Supplementary Results and Discussion, Fig. S20). In the bound conformation, by contrast, the turret moves toward the S4 voltage sensor and, consequently, the S3–S4 loop shifts laterally by 5 Å. This movement immobilizes the S4 voltage sensor in its activated-UP conformation (Fig. 2), causing thereby the hyperpolarizing shift of the $G_p$-$V_c$ relation and the underlying preferential stabilization of the open state (Figs. 1 and 9, and Movie #1). From this scenario, the SAS triad of Kv3.1 and Kv3.2 emerge as the determining player of the selectivity and mechanism of action for AUT5. The triad at the equivalent location in Kv3.4 is RGN, where the central glycine disrupts the transition to a helix-like structure in that region of the turret. Therefore, the orchestrated cascade of conformational changes described above cannot occur, explaining the relatively AUT5-insensitive phenotype of Kv3.4 (Fig. 9). This also explains why the single Kv3.2 A5G mutation and the triple Kv3.2 SAS/RGN mutation are so effective at neutralizing the positive modulation induced by AUT5 (Fig. 8). Supporting these explanations, MD simulations demonstrate how Kv3.1 turret mutations that introduce Kv3.4 turret residues increase the flexibility of the S1-S2 loop and disrupt the critical coupling between the S3-S4 loop and the turret and, therefore, would be expected to neutralize the positive modulation by AUT5 (Supplementary Results and Discussion, Fig. S20). In addition, docking of AUT5 against the simulated mutant channel found solutions that largely resemble the bound state of the compound. Therefore, the

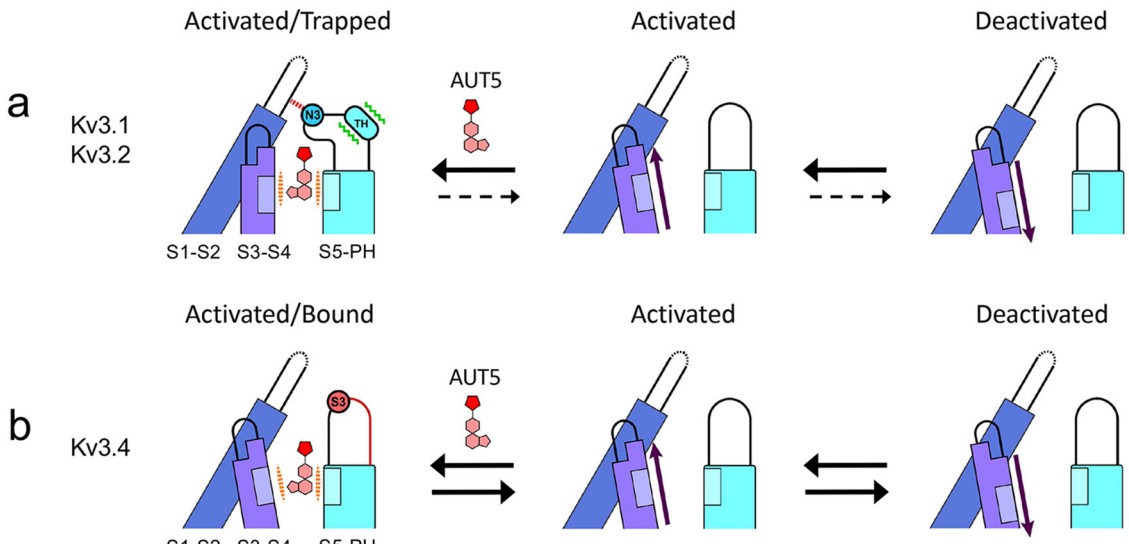

**Fig. 9 | The extracellular turret region of Kv3 channels governs the sensitivity to modulation by AUT5 and the mechanism of action.** For simplicity, this schematic only represents deactivation interactions that may take place between two neighboring subunits of the tetrameric domain-swapped Kv3 channel assembly. Our data suggest that positive modulation results from positive cooperativity involving four equivalent sites. **a** The deactivation pathway of Kv3.1 or Kv3.2 in the presence of AUT5. Upon binding of AUT5 in the conserved pocket, the turrets of Kv3.1 and Kv3.2 undergo a partial conversion to an alpha helix-like structure and fold over the bound AUT5 to trap it in the pocket. Consequently, the turret establishes new interactions with the S1–S2 and the S3–S4 loop to immobilize the S4 voltage sensor in its activated-UP conformation causing a slowing of

deactivation and, therefore, the channel's open state is preferentially stabilized. N3 represents the turret asparagine that is within atomic distance from the S1-S2 loop in the bound conformation of Kv3.1. **b** The deactivation pathway of Kv3.4 in the presence of AUT5. Since the binding site determinants are conserved in all Kv3 channels, AUT5 may occupy its pocket in Kv3.4. However, this binding results in no positive modulation because the Kv3.4 turret cannot undergo the critical rearrangements of its secondary and tertiary structure to stabilize the activated conformation of the VSD. Therefore, deactivation does not change in the presence of AUT5. S3 represents the Kv3.4 turret serine that occupies the equivalent N3 position in Kv3.1 and Kv3.2.

mutation-induced structural rearrangements impact the coupling between the turret and the S3–S4 loop but do not have a major impact on the abilities of AUT1 and AUT5 to occupy the binding site.

The importance of the S1–S2 loop is experimentally supported by the Kv3.2 chimeric construct hosting the S1–S2 loop of Kv3.4 (Kv3.4xKv3.2/S1-S2), which displayed significantly reduced positive modulation by AUT5 (Supplementary Information, Fig. S14). It is also notable that, whereas the S3–S4 loop is highly conserved among Kv3 channels, the S1–S2 loop is significantly divergent (Supplementary Information, Fig. S9). This is particularly remarkable between variants that have almost identical binding sites, S3–S4 loops and turrets, but divergent S1–S2 loops, such as Kv3.1 and Kv3.2. This difference helps explain why Kv3.2 is more sensitive to modulation by AUT5 than Kv3.1 and supports the role of the S1–S2 loop as the modulator of the critical interaction between the S3–S4 loop and the turret (Supplementary Results and Discussion). Although follow-up work is needed to sharpen the details of the proposed transduction mechanism, we have provided conclusive structural evidence for the presence of a modulator binding site in Kv3 channels. Moreover, we also determined with structural precision how the unique turret region outside the AUT5 binding site is ultimately responsible for the high selectivity of AUT5 for Kv3.1 and Kv3.2 and for triggering the mechanism that underlies the positive modulation.

While our manuscript was being considered for publication, Chen et al. reported the cryo-EM structure of a binding site for a distinct chemical class of positive modulators of Kv3.1 channels[65]. A direct overlay of the structures reported here with the structure reported by Chen et al. demonstrates nearly identical results (Supplementary Information, Fig. S21), which, together with our additional results, strongly supports the conclusions drawn here. It is remarkable that two distinct chemical classes of positive modulators occupy the same site and induce similar conformational changes, which broadens the

appeal of this unique site for the customization of Kv3 modulators with therapeutic potential.

## Methods
### Ethics
*Xenopus laevis* surgeries were performed according to a protocol approved by the Thomas Jefferson University Institutional Animal Care and Use Committee, #00001-2 (IACUC).

### Experimental design
To elucidate the mechanism of action of AUT5, we designed an orthogonal approach that combined (1) electrophysiological analysis to assess specificity among Kv channels and identify the biophysical basis of the mechanism of action; (2) cryo-EM to directly visualize the binding site of AU5 and AUT1; (3) systematic mutational and functional analyses to validate the critical molecular determinants of the mechanism of action; and (4) computational analyses (blind and focused docking, and MD simulations) to independently identify the unique AUT5 binding site and other potentially relevant interactions.

### cDNAs and site-directed mutagenesis
All human Kv3.2, Kv3.4a, and rat Kv1.2, Kv2.1, Kv3.1, Kv4.2, as well as *Drosophila* K-Shaw-F335A cDNAs encoding the investigated Kv channels were maintained in appropriate expression vectors suitable for in vitro transcription[66]. Site-directed mutagenesis was conducted according to the QuickChange protocol (Stratagene, La Jolla, CA). All chimeras were created by Phusion-based PCR (Thermofisher Scientific), and the primers used are listed in Supplementary Materials, Table S3[67]. All mutations were verified by automated DNA sequencing (Genewiz, South Plainfield, NJ). The mRNAs were synthesized with an in vitro RNA transcription kit (mMESSAGE mMACHINE by Ambion, ThermoFisher Scientific) followed by purification before heterologous expression in *Xenopus* oocytes[66].

## Reagents

AUT5 ((5 R)-5-ethyl-3-[6-(spiro[1-benzofuran-3,1'-cyclopropan]-4-yloxy)-3-pyridinyl]-2,4-imidazolidinedione; PubChem CID: 57410333) and AUT1 ((5 R)-5-Ethyl-3-(6-{[4-methyl-3-(methyloxy)phenyl]oxy}i-3-pyridinyl)-2,4-imidazolidinedione); PubChem CID: 53230344) were synthesized and purified by Autifony Therapeutics, LTD (Stevenage, UK)[17]. Lyophilized compounds were dissolved in DMSO to make a 10 mM stock, which was kept at 4 °C and was stable for up to 3 months. Immediately before use, AUT5 was diluted to the desired working concentrations in ND96 containing (in mM): 96 NaCl, 2 KCl, 1.8 CaCl$_2$, 1 MgCl$_2$, 5 HEPES, 2.5 sodium pyruvate, adjusted to pH 7.4 with NaOH. Collagenase A was purchased from Sigma (St. Louis, MO), Leibovitz's medium was purchased from Gibco (Thermo Fisher Scientific, Waltham, MA), PMA, DMSO and all other standard chemicals were purchased from Sigma-Aldrich (St. Louis, MO) (Supplementary Materials, Table S5).

## Heterologous expression of Kv channels

A standard collagenase-based dissociation technique was used to harvest mature stage V–VI oocytes suitable for heterologous expression and electrophysiological recording[66]. Ovaries were digested with collagenase A in calcium-free ND96 (in mM: 96 NaCl, 2 KCl, 1 MgCl$_2$, 5 HEPES, 2.5 sodium pyruvate, adjusted to pH 7.4 with NaOH) for 1.5 h. To improve yield, this step was repeated once with fresh collagenase A. Upon completion of the dissociation steps, the oocytes were washed at least 3 times with regular ND96, and at least 3 times additionally with Leibovitz's L-15 medium (500 mL Leibovitz's L-15 medium plus 220 mL H$_2$O, supplemented with 10 mM HEPES and 0.01 mg/mL gentamicin, and titrated with NaOH, pH 7.4). Isolated oocytes were then transferred to a 19 °C incubator and maintained in 35 mm Petri dishes containing Leibovitz's L-15. Mature oocytes lacking the follicular layer of cells (stage V–VI) were then selected for mRNA injection with a nanoliter microinjector using 3-000-203-G/X glass needles (Drummond Scientific Company)[66]. Typically, 46–92 nL of mRNA were injected per oocyte. The mRNA concentration was adjusted to obtain expression levels that are appropriate for TEVC (e.g., 2–7 µA at +50 mV, see below). Injected oocytes were maintained at 19 °C in Leibovitz's L-15 medium until they were transferred to the TEVC chamber for the recording of the expressed currents 24–72 h post-injection.

## Two-electrode voltage-clamping (TEVC)

Oocytes were transferred to a recording chamber containing ND96 (RC-3Z; Warner Instruments) and whole-oocyte currents were recorded at room temperature (21–23 °C) under TEVC conditions (OC-725C, Warner Instrument, Hamden, CT) according to established procedures[66]. The electrodes were filled with 3 M KCl and all recordings were conducted with ND96 in the bath. Freshly made AUT5 at the desired concentration was delivered into the recording chamber by means of a gravity-driven perfusion system with an exchange time of about 1 s and, during recording, the bath was continuously perfused. Data acquisition was performed using the Digidata 1440 A and running pClamp 9.2 and 10.3 (Molecular Devices, Sunnyvale, CA). Passive leak and capacitance transients were subtracted on-line by means of a standard p/4 protocol[66]. Specific voltage clamping protocols are described in the pertinent figures and figure legends. Currents were generally low-pass filtered at 1 kHz and digitized at 200–500 µs/sample point, except currents in Fig. 1c, which were low-pass filtered at 10 kHz and digitized at 20 µs/sample point.

## Data processing and analysis

Data processing, analysis, plotting and curve-fitting were performed in Clampfit 10.3. (Molecular Devices, Sunnyvale, CA) and Origin 9.1 Pro (OriginLab, Northampton, MA). Assuming the independently determined reversal potential ($V_r = -95$ mV), the peak chord conductance ($G_p = I_p/[V_c − V_r]$) was determined to characterize the voltage dependence of the expressed Kv channels. $I_p$ is the peak current evoked by

the command voltage $V_c$. The activation parameters were derived from the best fit to the $G_p$-$V_c$ relation assuming the following Boltzmann equation:

$$G_p(V_c) = \frac{G_{pmax}}{1 + e^{\frac{(V_c − V_{0.5})}{k}}} \tag{1}$$

Where $G_{pmax}$ is the maximum peak conductance, $V_{0.5}$ is the midpoint voltage, and $k$ is the slope factor. Assuming T = 22.5 °C, the equivalent gating charge z was estimated as follows: $z = 25.5/k$. To compare results from different oocytes and across different experiments, we calculated relative $G_p$ ($G_p/G_{pmax}$). $G_{pmax}$ was estimated from the best-fit Boltzmann equation. All $G_p$-$V_c$ data were obtained from paired sets (same oocyte, before and after application of AUT5). Measurement of the control $G_p$-$V_c$ relation was followed by determination of a stable response to AUT5 (current evoked by a depolarizing step to +60 mV, before and after bath application of AUT5) and subsequent measurement of the $G_p$-$V_c$ relation in the presence of AUT5. To assess modulation by AUT5, we quantified the changes of the $V_{0.5}$ ($\Delta V_{0.5}$), z ($\Delta z$) and $G_{pmax}$ (%$G_{pmax}$). The data were normalized with respect to the control in the absence of the compound. To evaluate whether a given mutation changed the sensitivity to modulation by the compound, the parameter changes induced by AUT5 for each mutant were compared to the AUT5-induced changes for the corresponding wild type.

To determine the time constants of current activation at various command voltages, we obtained the best fit to the rising phase of the currents (excluding the short initial current lag) assuming a 1st-order exponential equation or a sum of exponential terms:

$$I(t) = A_1 e^{−\frac{t}{\tau_1}} + A_2 e^{−\frac{t}{\tau_2}} + \dots + A_n e^{−\frac{t}{\tau_n}} \tag{2}$$

Where $A_1$, $A_2$ and $A_n$ are $\tau_1$, $\tau_2$ and $\tau_n$ are the amplitudes and time constants of the corresponding terms in the equation, respectively. Typically, no more than three terms were necessary to obtain a satisfactory fit. A similar approach was used to obtain the time constants of current deactivation at various tail command voltages. The weighted averages of the time constants ($\tau_W$, Eq. 3) were computed whenever two or more terms were necessary to obtain the best empirical description of the current trajectories, which was often the case at depolarized voltages.

$$\tau_W = \frac{A_1 \tau_1 + A_2 \tau_2 + \dots + A_n \tau_n}{A_1 + A_2 + \dots + A_n} \tag{3}$$

The $\tau_W$ was then plotted against the deactivation and activation command voltages to characterize the voltage dependence.

The concentration dependence of the AUT5-induced positive modulation (hyperpolarizing shift of the $G_p$-$V_c$ relation) was quantified by plotting $\Delta V_{0.5}$ vs. [AUT5]. Then, the EC$_{50}$ and Hill coefficient ($n_H$) were determined from the best fit of Eq. (4) shown below (X = [AUT5]).

$$Y(X) = Y_{MAX} \frac{X^{n_H}}{EC_{50} + X^{n_H}} \tag{4}$$

Where $Y_{MAX}$ is the maximal effect on the measured parameter. The fit was weighted assuming $w_i = 1/y_i$.

Statistical analysis.

All experiments were repeated with multiple independent batches of oocytes (at least two batches) and number of oocytes was shown as n in the figures. For descriptive purposes within the main text, numerical results are expressed as mean ± SEM, unless indicated otherwise. The paired Student $t$-test (two-tail) was used to evaluate $V_{0.5}$, z and $G_{pmax}$ changes induced by AUT1 and AUT5 (comparing before vs. after application of the compound). One-way ANOVA was

used to evaluate the differences between the two groups. However, since aggregated data from wild-type and mutants across different batches of oocytes were often not normally distributed and had unequal variances (based on the Levene's test), the non-parametric Kruskal-Wallis test was used to evaluate activation parameter changes induced by AUT5 ($\Delta V_{0.5}$, $\Delta z$ and $\%G_{pmax}$). When evaluating mutation-induced changes, the parameters derived from each mutant were individually compared to the parameters of the corresponding wild type.

### Expression and purification of human Kv3.1 channels

Kv3.1 was recombinantly expressed with BacMam expression system in Expi293F cells (Thermo Fisher Scientific, catalog # A14527) and purified as detailed in Chi, et al.[42]. Full-length human Kv3.1 (isoform A) cloned into LIC-adapted pHTBV C-terminally tagged Strep-II/10-His/GFP vector was used to generate Baculoviral DNA, which was then used to transfect Sf9 cells grown in Sf-900TM II media supplemented with 2% foetal bovine serum (Thermo Fisher Scientific). The resulting virus was further amplified by transducing Sf9 cells followed by incubation on an orbital shaker at 27 °C for 70 h, followed by harvesting by centrifugation at $900 \times g$. An Expi293F cell culture in mid-log phase ($2 \times 10^6$ cells/mL) in Freestyle 293 Expression Medium (Thermo Fisher Scientific) was infected with high-titre baculovirus (3% v/v) in the presence of 5 mM sodium butyrate. Cells were grown in orbital shaker at 37 °C with 8% $CO_2$ for 38 h before being harvested by centrifugation at $900 \times g$ for 10 min. The pelleted cells were washed with phosphate-buffered saline, pelleted again, then flash-frozen in liquid nitrogen (LN2) for storage in −80 °C freezer. Whole-cell pellet expressing Kv3.1 was resuspended to a total volume of 50 mL per 15 g cell pellet with buffer A (20 mM HEPES pH 7.5, 100 mM NaCl, 50 mM KCl) supplemented with 0.7% w/v lauryl maltoside neopentyl glycol (LMNG; Generon) and 0.07% cholesteryl hemisuccinate (CHS; Generon) for solubilization. The cells were solubilised at 4 °C for 1 h with gentle rotation, then centrifuged at $45,000 \times g$ for 1 h. Washed Strep-Tactin Superflow (IBA) was added to the lysate to a ratio of 0.4 mL resin per 100 mL lysate, and the slurry was gently rotated at 4 °C for 1 h. The resin was collected on a gravity flow column and washed with buffer B (buffer A with 0.003% LMNG and 0.0003% CHS), then with buffer B supplemented with 2 mM ATP and 5 mM $MgCl_2$. Kv3.1 was eluted with 6 CV of buffer B supplemented with 5 mM D-desthiobiotin followed by overnight tag-cleavage with TEV protease. The sample was then concentrated and subjected to a size-exclusion chromatography pre-equilibrated with buffer C (buffer A supplemented with 0.04% digitonin (Apollo Scientific)). After size exclusion chromatography, peak fractions were pooled and 1 mM EDTA was added. DMSO-solubilised stocks of either 100 mM AUT1 or 100 mM AUT5 were added to the samples to final concentrations of 300 μM. The samples were then incubated on ice overnight. The following day, the samples were concentrated to 60 μM for Kv3.1/AUT1 and 200 μM for Kv3.1/AUT5.

### Cryo-electron microscopy sample preparation, data collection, and data processing

All samples were frozen on Quantifoil Au R1.2/1.3 200-mesh grids freshly glow-discharged for 60 s. with plunge freezing performed on Vitrobot Mark IV (Thermo Fisher Scientific) set to 100% humidity, 4 °C, 30 s wait time and 3.5–15.0 s blotting time. Both Kv3.1/AUT1 and Kv3.1/AUT5 datasets were collected with EPU on a Titan Krios (Thermo Fisher Scientific) operating at 300 keV at eBIC (Harwell, UK). 14,344 (Kv3.1/AUT1) and 17,329 (Kv3.1/AUT5) super-resolution dose-fractioned micrographs with two times binning were collected on a K3 (Gatan) detector at 105,000x nominal magnification (pixel size = 0.831 Å/pix for Kv3.1/AUT1, 0.825 Å/pix for Kv3.1/AUT5). Micrographs were motion-corrected with Path Motion Correction on Cryosparc 3.3.1 (Structura Biotechnology)[68], and defocus values were estimated with Patch CTF function on Cryosparc. Particles

were initially picked with the Blob Picker function, and extracted particles were subjected to multiple rounds of 2D classification. Polished particles representing Kv3.1 were used as inputs to generate a Topaz-generated particle-picking model[69], followed by Topaz extraction function to pick particles. Particles extracted from this were subjected to 2D classifications. Polished particles from both blob picking and Topaz picking were pooled and duplicates were discarded. 50,000 randomly selected particles were used to generate ab initio models on Cryosparc, then all particles were subjected to non-uniform refinements with C1 and C4 symmetries imposed. After manual inspection of the reconstructed maps, particles with C4-refinement were further polished with local motion correction and CTF refinement functions on Cryosparc, followed by non-uniform refinement with C4 symmetry. These yielded 3D reconstructions of Kv3.1 with nominal resolutions of 2.9 Å for Kv3.1/AUT1 dataset and 2.5 Å for Kv3.1/AUT5 dataset, respectively.

### Model building and refinement of the structures

Structure model of human Kv3.1 (PDB ID: 7PHI) was fitted to the Kv3.1 maps on UCSF Chimera[70]. Structure models and restraints files for AUT1 and AUT5 were generated with AceDRG on CCP4i-cloud[71,72], which were then fitted to the cryo-EM maps. The models were refined with iterative cycles of manual building on Coot[73] and real space refine function on Phenix[74]. The models' geometries were verified in Phenix with MolProbity[75].

### Blind docking calculations

AUT5 structure file was obtained from PubChem (PubChem ID: 57410333)[76] and converted to a pdb file using OpenBabel 3.0.0[77]. A set of Kv channel structures was used as input for docking calculations. High-resolution structures for the AUT5-sensitive channel Kv3.1 were obtained directly from Protein Data Bank (IDs: 7PHH, 7PHI, 7PHK and 7PHL)[42]. Three-dimensional structures for Kv3.2 were generated using ColabFold[78], with default parameters (msa_mode: MMseqs2 (UniRef +Environmental); pair_mode: unpaired+paired; model_type: auto). AUT5-insensitive channel structures were obtained from PDB: Kv1-Kv2 Chimera (IDs: 2R9R, 6EBK, 7SIT)[79–81], Kv1.2 (ID: 2A79)[79] and Kv4.2 (ID: 7E84)[82]. For blind docking calculation, all Kv channel had their intracellular domains removed.

Docking was performed with Autodock Vina 1.1.2[83]. Docking solutions were resolved with an exhaustiveness parameter of 350, by searching a box volume of $120 \times 120 \times 90$ Å$^3$ containing the transmembrane domain of the protein receptor. AUT5 was allowed to have flexible bonds for all calculations. Clustering of docking solutions was carried out following a maximum neighborhood approach, with a proximity criterion (RMSD < 0.5 Å)[84,85]. When clustered, the solution contacts were merged, and the lowest energy solution was adopted as the bound ligand position. Among all suggested sites in the blind docking procedure, the site that had the highest percentage of turret residues in its composition was selected as the AUT5 binding site.

### Focused docking calculations

Focused docking calculations of AUT1 and AUT5 were performed with Autodock Vina 1.1.2[83] by taking into consideration the cryo-EM structures of the ligand-protein complex described herein. Docking solutions were resolved with an exhaustiveness parameter of 350, by searching a cubic box volume of $20 \times 20 \times 20$ Å$^3$ centered at the experimentally determined binding site of the molecule. AUT1 and AUT5 were allowed to have flexible bonds for all calculations. A minimum root-mean-square deviation (rmsd) between docking poses and the cryo-EM bound configuration of AUT1 and AUT5 was considered as the structural criterium to select docking solutions best reproducing the bound state of the compounds. Best docking solutions were selected for further comparative analysis of interaction energies.

## Reporting summary

Further information on research design is available in the Nature Portfolio Reporting Summary linked to this article.

## Data availability

The data that support this study are available from the corresponding authors upon request. The cryo-EM maps of the human Kv3.1a structures and the corresponding atomic coordinates have been deposited in the Electron Microscopy Data Bank under the accession codes EMD-18659 (AUT1-bound) and EMD-18660 (AUT5-bound); and the Protein Data Bank under the accession codes 8QUC (AUT1-bound); and 8QUD (AUT5-bound). Previously published structures can be found at EMD-13417 and 7PHI. The source data for electrophysiology experiments (Figs. 1, 4, 5, 7, 8 and Supplementary Fig. S1–S5, S13–S19) are provided as a Source Data file with this paper. Cryo-EM data and validation information (Fig. 2 and Supplementary Fig. S6; Table S1) are available from the Electron Microscopy Data Bank links provided above. Data files and models concerning docking calculations (Fig. 3, Supplementary Figs. S11–S12) and molecular dynamics simulations (Supplementary Fig. S20) are available from the Zenodo data repository [https://zenodo.org/records/10668705]. Source data are provided with this paper.

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

## Acknowledgements

We thank present and past members of the Covarrubias lab and the Jefferson Synaptic Biology Center on their helpful feedback during the execution of this work. This work was supported by a grant from Autifony Therapeutics, Ltd. (M.C.) and intramural funds from the Jefferson Synaptic Biology Center (M.C.); and grants from the National Council for Scientific and Technological Development CNPq (grants 02089/2019-5 and 200114/2020-4) (W.T.). Additionally, we received funding from the Wellcome Trust (grant No. 106169/Z/14/Z) and the Innovative Medicines Initiative 2 Joint Undertaking (JU) under grants agreements No. 875510 (EUbOPEN) (G.C., D.B.S.). The JU also receives support from the European Union's Horizon 2020 research and innovation programme and EFPIA and Ontario Institute for Cancer Research, Royal Institution for the Advancement of Learning, McGill University, Kungliga Tekniska Hoegskolan, and Diamond Light Source Ltd. We also acknowledge Diamond for access and support of the cryo-EM facilities at the UK national electron Bio-Imaging Centre (eBIC), proposal BI28713, funded by the Wellcome Trust, Medical Research Council, UK and Biotechnology Biological Sciences Research Council, UK. Oxford Particle Imaging Centre was funded by a Wellcome Trust JIF award (Grant No. 060208/Z/00/Z) and is supported by equipment grants from Wellcome Trust (093305/Z/10/Z). Publication made possible in part by support from the Thomas Jefferson University Open Access Fund.

## Author contributions

C.H.L., G.A., M.J.G., and M.C. conceived the study. G.A and A.M. designed and synthesized the compounds. Q.L., L.Z., N.P. and M.C. designed, performed, and analyzed the mutational and electrophysiological experiments. G.C. and D.B.S. designed, performed, and analyzed the cryo-EM experiments. L.C. and W.T. designed, performed, and analyzed the blind docking calculations and molecular dynamics simulations. Q.L., G.C., L.C., W.T., and M.C. created the visualizations. C.H.L., and M.C. obtained the funding. C.H.L., M.J.G., and M.C. were responsible for the administration and supervision of the project. M.C. wrote the manuscript and all authors contributed to reviewing and editing.

## Competing interests

A.M., N.P., G.A., M.J.G. and C.H.L. are employees and shareholders of Autifony Therapeutics, Ltd. Q.L., L.Z. and M.C. received funding from Autifony Therapeutics, Ltd. to conduct this work. The remaining authors declare no competing interests.
