## [Peer Review File · Nature Communications]

The binding and mechanism of a positive allosteric modulator of Kv3 channelsReviewer #1 (Remarks to the Author):

Kv channels are fundamental for the generation and propagation of electrical signals in most physiological processes. Reduced function of Kv3 has been indicated in multiple neurological diseases, making the discovery and characterization of potentiators very important. The authors here characterized how AUTs bind to and selectively potentiates subtypes of Kv3s. Comparing Kv3.1/3.2 that are sensitive to AUT5 potentiation and Kv3.4 that is not sensitive in mutagenesis studies, combining with structures and simulation, the authors identified bindings sites and potentiation mechanism through the turret. The binding site coincides with that of a different class of compounds recently reported from independent group, further demonstrating the relevance of this site in functional regulation. Overall, this work represents a significant progress toward the understanding of ligand regulation of Kv3 channels, with detailed analysis, and should be published after addressing a few minor points.

Minor points:

1. Fig. 1B. Is AUT5 normalized to the same G_{max}? Why the last data point (40mV) is missing? Are these single measurements? If these are averages, proper error bars should be shown. Is the difference in z before and after washout significant?
2. Fig. 4. Clearly mutations near the binding site have additional effects other than (in addition to) change of AUT binding affinities. Titration of AUT5 (like in Fig. 1G) with each mutation is preferable to separate changes in binding affinities from other effects. In addition, the roughly negative correlation between mutant-induced and AUT induced V_{mid} shift is interesting – why is this the case? Are mutations mimicking AUT-bound or non-bound state?
3. Fig. 5 referred to voltage protocol shown in Fig. 1, which is for tail current measurements. Are these tail currents? The same confusion for Fig. 1 referring to Fig. S1 for step protocols – representation did not include the number of steps, which seemed different for different channels as indicated in plots. Showing explicitly voltage protocol used for each (group) recording in the same main figure would help.
4. Fig. 8. Are these V_{0.5} changes calculated with respect to WT without AUT5, or corresponding mutants without AUT5?
5. Comparing between AUT and the recently reported Kv3.1 potentiator in the same / similar binding pocket (<https://doi.org/10.1073/pnas.2220029120>) and discussing possible similarities/differences in potentiation mechanism might expand the scope of the site identified here.

Reviewer #2 (Remarks to the Author):

This is a remarkably thorough and complete manuscript exploring the structural basis for modulation of Kv3 channels by two selective positive allosteric modulators. The authors present a no-stone-unturned investigation demonstrating that these compounds selectively stabilize the open state of Kv3.1 and Kv3.2, that they bind in a cavity between the voltage-sensing and pore domains and how rearrangements in the turret region are critically involved in the actions of the compounds as well as the selectivity towards Kv3.1 and Kv3.2 channels. I can't recall reviewing a manuscript recently where the work was so thoughtfully presented that I have effectively no concerns. The functional aspects of the work are particularly detailed in their depth and breadth, but I found all aspects of the study to be quite convincing. The authors are to be congratulated on putting together a superb study, and one

that Nature Comm should be proud to publish. The following are a few minor comments for the authors to consider as they revise their manuscript.

1) Recent structures of the Shaker Kv channel and the mechanism of slow inactivation might be worth noting in the discussion. The turret in that channel is very important for inactivation and the recent structures nicely explain earlier functional studies on the turrets role in slow inactivation (PMC8932672). The turret is also targeted by a nanobody that effectively inhibits Kv1.3 channels (PMC9253088), supporting the conclusions in this study about its importance.

2) In the discussion the authors mention ref 61 (PMC10589703) and list the ways in which the present study is more extensive and thorough. While I completely agree with those statements, it comes off as a bit defensive and unnecessarily demeaning of that study. The value and quality of the present manuscript should not be in question and once the editors have agreed that the study should be published in Nature Comm, I would suggest that the authors tone down this part of the discussion and simply point out how a structurally distinct allosteric modulator has been shown to bind to a similar region of Kv3.1, supporting the generality of the present findings.

3) Its clear to me that the G-Vs must have been normalized to Gmax in control solutions but it would be good to say that explicitly in the legends, where appropriate.

Authors' Response

We thank the reviewers for their positive review and helpful feedback. In this response, we are addressing their comments point-by-point (*indented and italicized text*). Also, we have accordingly highlighted text changes in the revised main manuscript text and Supplementary Information.

Reviewer #1 (Remarks to the Author):

Kv channels are fundamental for the generation and propagation of electrical signals in most physiological processes. Reduced function of Kv3 has been indicated in multiple neurological diseases, making the discovery and characterization of potentiators very important. The authors here characterized how AUTs bind to and selectively potentiates subtypes of Kv3s. Comparing Kv3.1/3.2 that are sensitive to AUT5 potentiation and Kv3.4 that is not sensitive in mutagenesis studies, combining with structures and simulation, the authors identified bindings sites and potentiation mechanism through the turret. The binding site coincides with that of a different class of compounds recently reported from independent group, further demonstrating the relevance of this site in functional regulation. Overall, this work represents a significant progress toward the understanding of ligand regulation of Kv3 channels, with detailed analysis, and should be published after addressing a few minor points.

We thank the reviewer for the positive appreciation of the work reported in our manuscript.

Minor points:

1. Fig. 1B. Is AUT5 normalized to the same G_{max} ? Why the last data point (40mV) is missing? Are these single measurements? If these are averages, proper error bars should be shown. Is the difference in z before and after washout significant?

- *Whenever we assessed the modulation by a compound, we normalized the G_p -V curve with respect to the G_{pmax} of the control (i.e., in the absence of compound). We have added this clarification to the Methods (lines 574-577) and figure legends (Figs. 1, 5 and 7).*
- *Thank you for catching this error. We missed the +40 mV data point when preparing the final graph. We have revised this graph in Fig. 1B, now including the data point at +40 mV.*
- *The data on Fig. 1B are means \pm SEM ($n=23$, as indicated on the graph). The SEM bars are about as big as the symbol size and are, therefore, nearly obscured. We have added this statement to the legend of Fig. 1, as we did for other similar plots on other figures.*
- *The gating parameters ($V_{0.5}$, z , and G_{max}) remained significantly different from control after washout. Demonstrating nearly complete reversibility, however, the G -V curve after washout is close to the control curve (Fig. 1B). It was generally difficult to reach 100% washout, suggesting the possible role of membrane partitioning of the amphiphilic compound.*

2. Fig. 4. Clearly mutations near the binding site have additional effects other than (in addition to) change of AUT binding affinities. Titration of AUT5 (like in Fig. 1G) with each mutation is preferable to separate changes in binding affinities from other effects. In addition, the roughly negative correlation between mutant-induced and AUT induced V_{mid} shift is interesting – why is this the case? Are mutations mimicking AUT-bound or non-bound state?

- *We agree with the reviewer. To functionally assess the effects of mutations on the apparent binding affinity of the compound, it would have been preferable to generate concentration-response curves (e.g., Fig. 1G). However, the tested mutations also affect voltage-dependent gating. This was expected, given that the mutated residues are lining a critical interface between the VSD and PD. We, therefore, concluded that the functional assay employed here would not allow an unambiguous separation of gating vs. binding effects. Generating the concentration/response curves would have been possible, albeit very laborious and time consuming because multiple concentrations could not be tested on the same oocyte. The compound washout/exchange is slow and not always entirely complete. Consequently, the risk of cumulative and drifting effects that could distort the concentration/response curves from individual oocytes is significant. Based on a risk/benefit assessment, we decided not to pursue the time-consuming task of generating complete concentration/response curves for every mutation tested.*

Nevertheless, selected mutations of residues lining the pocket impact the modulation in a manner that is not entirely due to altered gating. Moreover, as pointed out by the reviewer, an apparent negative correlation between the effect of mutations on compound dependent modulation and voltage dependence is very interesting because it suggests that a possible steric effect of the mutated amino acid side chain in the binding pocket may mimic compound occupancy and the resulting positive modulation. This is noted in the manuscript (lines 234-240 and lines 440-445). Therefore, the effects of mutations on gating and compound binding may not be separable. Further systematic analysis to test different amino acid side chains at each location would be necessary to explore this possibility further. This was, however, outside of the scope of this already very extensive study.

3. Fig. 5 referred to voltage protocol shown in Fig. 1, which is for tail current measurements. Are these tail currents? The same confusion for Fig. 1 referring to Fig. S1 for step protocols – representation did not include the number of steps, which seemed different for different channels as indicated in plots. Showing explicitly voltage protocol used for each (group) recording in the same main figure would help.

Thank you for identifying the error. It should be the protocol shown on Fig. S1D. We will show or describe the voltage protocols in detail wherever necessary (e.g., Fig. S1D legend).

4. Fig. 8. Are these $V_{0.5}$ changes calculated with respect to WT without AUT5, or corresponding mutants without AUT5?

For every mutant, the $\Delta V_{0.5}$ values are calculated with respect to the control in the absence of AUT5. Please see Figs. S18 and S19, which show the effects before and after AUT5. The paired data points on these graphs were used to compute the $\Delta V_{0.5}$ induced by AUT5 for each mutant, and this was compared to the AUT5-induced $\Delta V_{0.5}$ for the WT. We are therefore assessing whether a mutation has changed the sensitivity to the modulation by AUT5, relative to the WT. We have clarified it explicitly in the revised manuscript (Materials and Methods; lines 575-577).

5. Comparing between AUT and the recently reported Kv3.1 potentiator in the same / similar binding pocket (<https://doi.org/10.1073/pnas.2220029120>) and discussing possible similarities/differences in potentiation mechanism might expand the scope of the site identified here.

Thank you for suggesting addition of a direct comparison between the structures reported in our manuscript and the structure published by the Merck group. We have added a new figure directly depicting this comparison to the Supplementary Information (Fig. S21). Also, we revised the last paragraph of the Discussion to note the remarkable similarity between the structures (lines 494-500).

Reviewer #2 (Remarks to the Author):

This is a remarkably thorough and complete manuscript exploring the structural basis for modulation of Kv3 channels by two selective positive allosteric modulators. The authors present a no-stone-unturned investigation demonstrating that these compounds selectively stabilize the open state of Kv3.1 and Kv3.2, that they bind in a cavity between the voltage-sensing and pore domains and how rearrangements in the turret region are critically involved in the actions of the compounds as well as the selectivity towards Kv3.1 and Kv3.2 channels. I can't recall reviewing a manuscript recently where the work was so thoughtfully presented that I have effectively no concerns. The functional aspects of the work are particularly detailed in their depth and breadth, but I found all aspects of the study to be quite convincing. The authors are to be congratulated on putting together a superb study, and one that Nature Comm should be proud to publish. The following are a few minor comments for the authors to consider as they revise their manuscript.

We thank the reviewer for the enthusiastic review.

1) Recent structures of the Shaker Kv channel and the mechanism of slow inactivation might be worth noting in the discussion. The turret in that channel is very important for inactivation and the recent structures nicely explain earlier functional studies on the turrets role in slow inactivation (PMC8932672). The turret is also targeted by a nanobody that effectively inhibits Kv1.3 channels (PMC9253088), supporting the conclusions in this study about its importance.

Thank you for suggesting discussion of recent studies that additionally highlight the functional and pharmacological importance of the Kv channel turret. We have noted and cited the suggested studies in the revised manuscript (lines 424-427).

2) In the discussion the authors mention ref 61 (PMC10589703) and list the ways in which the present study is more extensive and thorough. While I completely agree with those statements, it comes off as a bit defensive and unnecessarily demeaning of that study. The value and quality of the present manuscript should not be in question and once the editors have agreed that the study should be published in Nature Comm, I would suggest that the authors tone down this part of the discussion and simply point out how a structurally distinct allosteric modulator has been shown to bind to a similar region of Kv3.1, supporting the generality of the present findings.

We agree to tone down the discussion of the recently published structure from the Merck group and apologize for coming off defensively about it. Under Supplementary Information, we have added a new figure to factually compare the two structures, as suggested by Rev. 1. The last paragraph of the Discussion has been accordingly revised (lines 494-500).

3) Its clear to me that the G-Vs must have been normalized to Gmax in control solutions but it would be good to say that explicitly in the legends, where appropriate.

We agree to more explicitly indicate how the G/V curves are normalized. As suggested (see Rev. 1 comments) and wherever necessary, we have added this information under Materials Methods (lines 578-581) and in the figure legends.

Reviewer #1 (Remarks to the Author):

The authors have addressed my concerns and I have no further comments. These findings should be shared with the community.

Reviewer #2 (Remarks to the Author):

The authors have done a nice job of revising the manuscript to address the suggestions of both reviewers, and in my view the study is now ready for publication. Congratulations once again on a beautiful piece of work!